



# Using hydrologic landscape classification and climatic time series to assess hydrologic vulnerability of the Western U.S. to climate

Chas E. Jones[1]*, Scott G. Leibowitz[2], Keith A. Sawicz[3], Randy L. Comeleo[2], Laurel E. Stratton[4], Phillip E. Morefield[5], Chris P. Weaver[6]

[1] Oak Ridge Institute for Science and Education (ORISE), c/o U.S. Environmental Protection
Agency, Center for Public Health and Environmental Assessment, Pacific Ecological Systems Division, 200 SW 35th
St., Corvallis, OR 97333, USA; Current affiliation: Affiliated Tribes of Northwest Indians, Corvallis, OR 97333, USA

[2] U.S. Environmental Protection Agency, Center for Public Health and Environmental Assessment, Pacific Ecological
Systems Division, 200 SW 35th St., Corvallis, OR 97333, USA

[3] Oak Ridge Institute for Science and Education (ORISE), c/o U.S. Environmental Protection
Agency, Center for Public Health and Environmental Assessment, Pacific Ecological Systems Division, 200 SW 35th
St., Corvallis, OR 97333, USA

[4] c/o U.S. Environmental Protection Agency, Center for Public Health and Environmental
Assessment, Pacific Ecological Systems Division, 200 SW 35th St., Corvallis, OR 97333, USA

[5] U.S. Environmental Protection Agency, Center for Public Health and Environmental Assessment, Health and
Environmental Effects Assessment Division, Washington, DC 20460, USA

[6] U.S. Environmental Protection Agency, Center for Public Health and Environmental Assessment, Health and
Environmental Effects Assessment Division, Research Triangle Park, NC 27709, USA

*Correspondence to:* Chas E. Jones (chas@chasjones.com)





**Abstract.** We apply the hydrologic landscapes (HL) concept to assess the hydrologic vulnerability of the western United States (U.S.) to projected climate conditions. Our goal is to understand the potential impacts for stakeholder-defined interests across large geographic areas. The basic assumption of the HL approach is that catchments that share similar physical and climatic characteristics are expected to have similar hydrologic characteristics. We map climate vulnerability by integrating the HL approach into a retrospective analysis of historical data to assess variability in future climate projections and hydrology, which includes temperature, precipitation, potential evapotranspiration, snow accumulation, climatic moisture, surplus water, and seasonality of water surplus. Projections that are not within two-standard deviations of the historical decadal average contribute to the vulnerability index for each metric. This allows stakeholders and/or water resource managers to understand the potential impacts of future conditions. In this paper, we present example assessments of hydrologic vulnerability of specific geographic locations (Sonoma Valley, Willamette Valley, and Mount Hood) that are important to the ski and wine industries to illustrate how our approach might be used by specific stakeholders. The resulting vulnerability maps show that temperature and potential evapotranspiration are consistently projected to have high vulnerability indices for the western U.S. Precipitation vulnerability is not as spatially uniform as temperature. The highest elevation areas with snow are projected to experience significant changes in snow accumulation. The seasonality vulnerability map shows that specific mountainous areas in the West are most prone to changes in seasonality, whereas many transitional terrains are moderately susceptible. This paper illustrates how the HL approach can help assess climatic and hydrologic vulnerability across large spatial scales. By combining the HL concept and climate vulnerability analyses, we provide a planning approach that could allow resource managers to consider how future climate conditions may impact important economic and conservation resources.

## 1 Introduction

A stable and predictable water supply is imperative to national security (National Intelligence Council, 2012), especially as it pertains to the global food supply, and the threats of increased flooding, droughts, wildfire, and more extreme temperatures (Mancosu et al., 2015; Mekonnen and Hoekstra, 2016). The recognition of the potential threats of climate on society is important, and the development of planning tools could help decision-makers assess the risk imposed by projected environmental changes, such as those imposed by climate, population growth, or habitat conversion (Glick et al., 2011; Lawler et al., 2010). Environmental changes related to climate and hydrology will not impact stakeholders equally across sectors, thus the specific concerns and adaptation strategies of different industries will vary.

Numerous studies have examined projected changes in climate and hydrology on regional and national scales that included the western U.S. The Third National Climate Assessment (http://nca2014.globalchange.gov) is a comprehensive resource for climate-related research in the U.S. (Melillo et al., 2014). Nolin and Daly (2006) mapped climate-related risk to snow-dominated areas and ski areas in the Pacific Northwest. Mote et al. (2005) compared the spatial patterns of snow water equivalent observations to model simulations in the western U.S. Brown and Mote (2009) examined projected changes in snow water equivalent globally based on 14 model projections. Barnett et al. (2005) identified potential climate-driven water supply deficits in snow-dominated areas around the globe, although rising water demands have been found to greatly outweigh potential climate impacts on future (year 2025) water



supply (Vorosmarty et al., 2000). McAfee (2013) examined projected changes in potential evapotranspiration (PET,
calculated using numerous methods) between 2002-2011 and 2079-2098. The findings are consistent across studies
in many areas of the globe including across the conterminous U.S., but other regional PET predictions were
inconsistent and sensitive to the method of calculation. Hill et al. (2013, 2014) predicted thermal vulnerability of
streams and river ecosystems to climate across the U.S., while Battin et al. (2007) found that in regards to salmon
habitat, snow-dominated streams were more vulnerable habitat than lowland streams. The analyses of Nijssen et al.
(2001) on hydrologic sensitivity of rivers globally found: 1) Ubiquitous warming, with greatest warming in winter
months at higher latitudes, 2) More precipitation with high variability, 3) Early to mid-spring snowmelt caused
increased spring streamflow peak in coldest basins, decreased spring runoff and increased winter runoff in transitional
basins, 4) Tropical or mid-latitude basins had decreased annual runoff, and 5) High latitude basins had increased
annual streamflow. In response to droughts of the recent past, Mann and Gleick (2015) highlight the strong correlation
between very hot years and very dry years; thus as temperatures increase, precipitation is becoming more scarce. A
study by Cook et al. (2015) found a growing risk of unprecedented drought in the western U.S. based on temperature
projections and no clear pattern in future precipitation.
"Vulnerability" has many accepted definitions depending upon discipline and application (Adger, 2006; Füssel, 2007).
Vulnerability assessments often integrate exposure, sensitivity, and adaptive capacity to stressors (Adger, 2006;
Füssel, 2007; Füssel and Klein, 2006; IPCC, 2014). Researchers have studied vulnerability at varying scales across
numerous regions for a diversity of stakeholders, and they tend to focus on the most relevant metrics for their particular
application (Farley et al., 2011; Glick et al., 2011; IPCC, 2014; Nolin and Daly, 2006; U.S. Global Change Research
Program, 2011; Watson et al., 2013). Yet, better products and services are needed to enable local communities to plan
for and respond to hydrologic change, which includes services that improve understanding, observing, forecasting,
and warning about significant hydrologic events (Tansel, 2013). Glick et al. (2011) and Lawler et al. (2010) both
emphasize the importance to managers of understanding the potential impacts of climate on the resources that they
manage.
There have been many efforts to assess hydrologic vulnerability related to specific stakeholders, ecosystems, or
locations. For example, Vörösmarty et al. (2000) examined the vulnerability of global water resources to changes in
climate and population growth. Hill et al. (2014) assessed stream temperature vulnerability to climate for sites across
the U.S. In another example, Winter (2000) suggested that the vulnerability of wetlands to changes in climate depends
upon their position within the hydrologic landscape.
There are opportunities to build upon previous efforts to map hydrologic vulnerability across large geographic areas,
while creating tools that stakeholders may use to understand the potential impacts for their asset of interest in specific
watersheds. Winter (2001) described the concept of classifying the physical landscape and climatic properties of
catchments based on hydrologic landscapes (HL). Surface and ground water availability in watersheds is impacted by
differences in geology, terrain, soils, seasonal temperature patterns, precipitation magnitude, and precipitation timing
(Tague et al., 2013; Winter, 2001) and are not uniform across regions (Hamlet, 2011; Jung and Chang, 2012; Tague
and Grant, 2004). Catchments that share similar key physical and climatic characteristics are expected to have similar



hydrologic characteristics; i.e., surface and ground water interactions, deposition, timing, and accumulation of
precipitation, surface runoff patterns, and groundwater flow (Nolin, 2011; Thompson and Wallace, 2001).
The HL concept has been applied to the U.S. (Wolock et al., 2004) and modified approaches have been used in Oregon
(Leibowitz et al., 2014; Patil et al., 2014; Wigington et al., 2013), Nevada (Maurer et al., 2004), the Pacific Northwest
(Comeleo et al., 2014; Leibowitz et al., 2016), and Bristol Bay, Alaska (Todd et al., 2017). In applying the HL
approach in Oregon and the Pacific Northwest, two climatic factors and three landscape characteristics were
categorized for each catchment; the resulting classification allows the prediction of catchment-scale hydrologic
behavior across large spatial scales. The approach shows promise in predicting seasonal and monthly hydrologic
patterns (Leibowitz et al., 2014). Leibowitz et al. (2014) adapted the classification system applied by Wigington et al.
(2013) to illustrate the applicability of HLs for representing normal (1971-2000) monthly average streamflow in three
case study watersheds in Oregon. They used climate projections (2041-2070) to estimate hydrologic behavior of
catchments relative to 1971-2000. Leibowitz et al. (2016) expanded the approach and applied the HL classification to
Oregon, Washington, and Idaho.
A number of tactics have been used to investigate the influence of climate on hydrologic behavior (Luce and Holden,
2009; Safeeq et al., 2014; Vano et al., 2015). To extend the work previously completed from HL-based climate
projections, we assess climate vulnerability at the catchment scale by integrating the HL approach into an analysis of
climatic variability. Our hydrologic landscape vulnerability analysis (HLVA) provides spatially continuous,
application-specific estimates of climatic vulnerability. One of the benefits of the HLVA is to place modern and
projected environmental changes in the context of available historic data. In the HLVA, we use proxies for the three
components of vulnerability: a) historic climate data and their derivatives as proxies for sensitivity; b) climate
projections as proxies for exposure; and c) qualitative considerations of ecosystems, stakeholders, or industries as
proxies for adaptive capacity. The HLVA assesses vulnerability to changes in temperature, precipitation, potential
evapotranspiration, snow accumulation, climatic moisture, surplus water, and seasonality of the water surplus. This
method highlights areas that are projected to experience deviations from historic conditions to understand the patterns
in magnitude, timing, and type of precipitation and the quantity and seasonality of available water at a catchment
scale.
We apply the HL concept with the goal of assessing the hydrologic vulnerability of the western U.S. to magnitude and
variability in climate projections. We analyzed this data to address three research objectives: 1) develop an index of
vulnerability based on past and projected climate behavior; 2) map areas that are projected to be more vulnerable to
environmental changes associated with climate; and 3) determine the vulnerability indices of seven metrics
(temperature, precipitation, snow accumulation, PET, surplus water (S'), Feddema Moisture Index (FMI; Feddema,
2005), and seasonality) for specific geographic areas, including three examples of industries that are economically
important in the region.
**2 Methods**
**2.1 Study Area**



The study area includes the states of Washington, Oregon, Idaho, California, Nevada, and Arizona in the western U.S.
(Fig. 1). These states extend across a wide range of climates and diverse physiographic settings. The lowest elevation
across the six states is 85 m below sea level (Death Valley, California), while the highest elevation is 4421 m above
sea level (Mt. Whitney, California) [U.S.G.S. National Elevation Dataset available at:
https://nationalmap.gov/elevation.html]. The Sierra-Nevada Mountains are oriented in a north-south direction near the
eastern border of California and transition to the Cascade mountain range that runs in a north/south direction through
Oregon and Washington. (US Topo Quadrangles available at: https://nationalmap.gov/ustopo). However, there are
numerous mountain ranges in each of the other states as well. The Sierra-Nevada and Cascade mountain ranges
generate orographic effects that cause upwind areas to the west to have much greater precipitation relative to the
downwind, eastern regions (Dettinger et al., 2004; Siler et al., 2013). High elevation areas receive most of their
precipitation as snow (Brekke et al., 2009; Mote et al., 2005), while lowland and coastal areas receive their
precipitation mostly as rain (Brekke et al., 2009; Mock, 1996), but much of the six-state area receives a balance of
snow and rain. The topographic differences across the landscape drive precipitation patterns across the six state study
area and cause large differences in the total annual precipitation or the seasonality of maximum precipitation (Mock,
1996). In the arid southwest, summer monsoons deliver most of the annual precipitation (Mock, 1996), whereas in the
Pacific Northwest, winter rains and snows are the dominant form of precipitation (Mock, 1996). However, the western
U.S. is regularly affected by atmospheric rivers that deliver large quantities of rain or snow over short periods
(Dettinger, 2011; Hidalgo et al., 2009). The seasonal variability of surface air temperature varies widely across the
study area. Portions of each state in our study area are classified as deserts with summer maximum temperatures
regularly exceeding 40°C (NOAA State Climate Extremes Committee, 2016). Each state in the study area has also
recorded temperatures less than -40°C (NOAA State Climate Extremes Committee, 2016). Some portions of the study
area have very mild climates with little seasonal variation in temperature (Daly, 2016b). Bedrock geology in the study
area varies from high permeability sedimentary deposits or relatively recent volcanic deposits, to low permeability
igneous metamorphic and sedimentary formations and older volcanics (Comeleo et al., 2014; Stratton et al., 2016).

**2.2 Hydrologic landscape classification**

The study area was divided into 29,356 assessment units (AUs). The AUs are aggregations of NHDPlusV2 catchments
(McKay et al., 2012) that were grouped to have a target area of 80 km$^2$, as described in Wigington et al. (2013) and
modified by Leibowitz et al. (2016). For this analysis, we retain an AU if its centroid was located within the boundary
of our project area or if the AU extended across an international boundary. All AU polygons are also clipped to the
international boundary of the U.S. These conditions allow us to avoid edge effects at international and state borders
by avoiding overlapping AUs at state boundaries and analyzing the HLs up to all international borders. The project
boundary was defined by merging these AUs into a single polygon.
Wigington et al. (2013) developed their HL classification based on climatic and physical characteristics of the physical
watershed. They defined five indices to characterize the major drivers that control the magnitude and timing of water
movement through the landscape and into the ground or stream network: (1) climate, which describes the overall
availability of water on the landscape, (2) seasonality of water surplus, which is the season when the maximum excess
of water is available to infiltrate into the soil column or flow as surficial runoff, (3) subsurface permeability, (4) terrain,





and (5) surface permeability. Note that Wigington et al. (2013) referred to subsurface and surface permeability as
aquifer and soil permeability, respectively. The five HL indices, described in more detail below (Sections 2.2.1 through
2.2.5), are typically concatenated into a 5-character HL code (e.g., WsLMH, SwHTH, or DfHfL) that characterizes an
AU.
Leibowitz et al. (2016) developed an HL map of the Pacific Northwest (PNW, consisting of Oregon, Idaho, and
Washington) based on a modification of the Wigington et al. (2013) approach (herein described as the modified
Wigington et al. (2013) approach). For the current effort, we used the modified Wigington et al. (2013) approach to
develop an HL classification of California, Nevada, and Arizona [referred to as the southwest]. This was then
combined with the PNW map (Leibowitz et al., 2016) to create an HL map of the six western states.

**2.2.1 Climate**

The Wigington et al. (2013) approach derived the climate index from the FMI (Feddema, 2005):

$$FMI = \begin{cases} 1 - \frac{PET}{P} & if \ P \geq PET \\ \frac{P}{PET} - 1 & if \ P < PET \end{cases} \tag{1}$$

where FMI (Eq. (1)) values range from -1.0 (arid) to 1.0 (very wet). P is the mean precipitation (mm) over a 30-year
normal, which is derived from climate data described in Section 2.3, and PET is the potential evapotranspiration (mm)
calculated using the Hamon (1961) method, that utilizes mean daily temperature, daytime length (calculated based on
latitude), and a calibration coefficient. The range of FMI values was the basis for a climate index consisting of six
classes: arid (A; $-1.0 \leq FMI < -0.66$), semiarid (S; $-0.66 \leq FMI < -0.33$), dry (D; $-0.33 \leq FMI < 0.0$), moist (M; $0.0 \leq$
FMI $< 0.33$), wet (W; $0.33 \leq FMI < 0.66$), and very wet (V; $0.66 \leq FMI < 1.0$) (Wigington et al., 2013). FMI was
calculated from regional precipitation rasters (described in Section 2.3) for each period of interest. The FMI value was
then averaged over each AU.

**2.2.2 Seasonality**

We used the Leibowitz et al. (2016) approach to develop a seasonality index that identifies the season of the maximum
monthly average snowpack-corrected surplus water (S'$_m$):

$$S'_m = S_m - \Delta PACK^*_m$$

$$= (P_m - PET_m) - (PACK^*_m - PACK^*_{m-1}) \tag{2}$$

where S'$_m$ (Eq. (2)) is the average snowpack-corrected water surplus (mm) for month $m$, S$_m$ is monthly water surplus
(P - PET), and P$_m$ and PET$_m$ are monthly precipitation and monthly PET, respectively. PACK$_m^*$ is a monthly bias-
corrected snowpack value (in mm of snow water equivalent, or SWE) restricted to values greater than zero, based on
the Leibowitz et al. (2016) modifications to the Leibowitz et al. (2012) snowpack model. Note, however, that
$\Delta PACK^*_m$ can have negative values, which represents snow melt. For each month, S'$_m$ was calculated for the regional
raster, before identifying the month of maximum S'$_m$ for the majority of pixels in each AU. The month of maximum
S'$_m$ was used to identify the season of maximum S'$_m$ based upon four seasonality classes: fall (f; October–December),
winter (w; January–March), spring (s; April–June), and summer (u; July–September). The PNW analysis by
Leibowitz et al. (2016) only included two seasonality classes; summer seasonality did not occur, while fall and winter
were combined into a winter class, since this represented the PNW's wet season.  For our analysis, we kept winter and



fall separate and used all four seasonality classes, because fall and winter are distinct seasons in other parts of the
nation.

**2.2.3 Subsurface permeability**

Leibowitz et al. (2016) utilized the Comeleo (2014) aquifer permeability dataset.  We applied a similar approach from
the Stratton et al. (2016) aquifer permeability datasets, which is herein referred to as subsurface permeability. Each of
these datasets classify the subsurface permeability into high (H) and low permeability (L) classes, which are assigned
with a threshold guideline of $8.5 \times 10^{-2}$ m day$^{-1}$ hydraulic conductivity. Using these data, we analyzed the subsurface
permeability of each AU by identifying the subsurface permeability class for the majority of pixels within each AU in
the three south western states.

**2.2.4 Terrain**

To classify terrain, we used the same approach as Wigington et al. (2013). We analyzed a 30 m Digital Elevation
Model to classify the landscape based upon the topographic characteristics of each AU. "Mountainous" (M) areas had
AUs with <10 % of the area identified as flat (< 1 % slope) and greater than 300 m of total relief. AUs with more than
50 % area having < 1 % slope were classified as "flat" (F).  All other AUs were identified as "transitional" (T).

**2.2.5 Surface permeability**

For surface permeability, the Wigington et al. (2013) HL approach utilized the STATSGO soil permeability raster
developed by Pennsylvania State University Center for Environmental Informatics (www.cei.psu.edu) for the top 10
cm of soil (Miller and White, 1998) in the conterminous U.S. The STATSGO soils database was selected because of
its complete coverage of the conterminous U.S., despite SSURGO's higher spatial resolution, which did not have
complete spatial coverage of the U.S. They identified whether the majority of each AU had high (H; >1.52 cm/hr) or
low (L; $\leq$ 1.52 cm h$^{-1}$) soil permeability. We applied the same approach to classify surface permeability of each AU
into two classes throughout the region.

**2.3 Climate analyses**

**2.3.1 Modern climate normal (1971−2000)**

Average monthly precipitation and mean temperature were acquired from Parameter-elevation Regressions on
Independent Slopes Model (PRISM; Daly, 2016b) data for our normal climatic period at a resolution of approximately
400 m. The PRISM Climate Mapping Program is an ongoing effort to produce detailed, spatial climate datasets (Daly,
2016a; Daly et al., 2000). PRISM uses point measurements of climate data and a digital elevation model to map
climate across the U.S. from 1895−present, including regions impacted by high mountains, rain shadows, temperature
inversions, coastal regions, and associated complex meso-scale climate processes. Using ArcGIS (ESRI, 2016), the
data were clipped to the project boundary and used to calculate the average  for our seven metrics (monthly
temperature, precipitation, PET,  surplus water, snow water equivalent, FMI, climate index, and seasonality of water
surplus) for the normal period. Each of these metrics are inputs to or products of the HL classification process.

**2.3.2 Historical climate analyses (1901−2010)**

Unlike with monthly precipitation and temperature data, a time series of gridded daily historical climate data at a
spatial resolution of 400 m was not available. Daily PRISM data is freely available at 4 km resolution, and this was
what we used to develop the historical climate analyses for the 1901−2010 period. Gridded data for daily mean



temperature and precipitation were clipped to the project boundary and averaged for each month over each decade
(i.e., 1901−1910, 1911−1920, etc.). The data were then statistically downscaled to 400 m using the delta method
(Hijmans et al., 2005; Ramirez-Villegas and Jarvis, 2010) to match the spatial resolution of the modern climate normal
data (using the 400 m resolution, monthly PRISM climate normal for 1971−2000 period as the high resolution dataset).
We acknowledge the inaccuracies and uncertainty imposed in the temperature and precipitation datasets by applying
the downscaling functions to the original climate projections, however since these 400 m resolution monthly averages
are normally distributed (Trzaska and Schnarr, 2014) and the data are to be aggregated to our 80 km$^2$ (on average)
AUs, the trade-offs were deemed acceptable and preferable for characterizing the hydrology and climate for these
analyses.
Using the approaches described herein, the downscaled data were used to calculate the average monthly PET, surplus
water, snow water equivalent, FMI, and seasonality of water surplus for each decade. Summary figures were generated
from this data depicting spatial distribution of climate and seasonality for each decade across the project area. These
data were compared to the modern climate normals using spatially continuous time series analyses.
**2.3.3 Future climate analyses (2041−2070)**
In order to explore the potential range of modeled climatic response for the study area, we selected ten climate model
projections from the full ensemble of World Climate Research Programme's Coupled Model Intercomparison Project
phase 5 multi-model ensemble climate dataset projections (WCRP CMIP5; http://cmip-pcmdi.llnl.gov/cmip5; Taylor
et al., 2012). These models are based on the Representative Concentration Pathway (RCP) 8.5 emissions scenario,
which assumes the highest rate of emissions into the 21$^{st}$ century. We only used this emissions scenario to reduce the
complexity of the analyses. To select the specific model simulations to use in this study, we created a scatterplot
comparing future temperature and precipitation change for the different CMIP5 models over the project area.  We
selected ten models that spanned the range of predicted climatic responses of the full ensemble (Fig. 2), including
drier, wetter, colder, and warmer responses. Average monthly precipitation and temperature for the ten projections
(Table 1) were acquired from the monthly Bias-Correction and Spatial Disaggregation (BCSD) archive (Bureau of
Reclamation, 2014) for the 2041−2070 period. These data were clipped to the project boundary and resampled to a
400 m grid using a bilinear approach (ESRI ArcGIS v10.4) to match the resolution and spatial extent of the modern
climate normal data. The average monthly PET, surplus water, snow water equivalent, FMI, and seasonality of water
surplus were calculated from the future climate data for each assessment unit. Summary figures were generated that
illustrate the spatial distribution of climate and seasonality for each climate projection. The differences in FMI and
seasonality of water surplus from the normal period were also mapped and compared.
**2.4 Mapping vulnerability indices**
As discussed in the introduction (Section 1), vulnerability can be measured by assessing the exposure, sensitivity, and
adaptive capacity of a system to change (Adger, 2006; Füssel, 2007; Füssel and Klein, 2006; IPCC, 2014). Historic
hydrology and climate are primary drivers for ecosystem change (Nelson, 2005), and are critical to certain industries
and stakeholders in particular areas; thus historic hydrology and climate serve as proxies for the sensitivity of those
systems to environmental change. In the assessment of hydrologic vulnerability, we evaluated the variability in
historical climate data and our derived hydrologic metrics as a proxy for sensitivity. Likewise, we used future climate



projections as a proxy for exposure to environmental change. Projections that fell outside of historic observations
should then be associated with increased levels of exposure. In terms of adaptive capacity, we assumed that the systems
present in a location are adapted to the historic observed variability in conditions. We also assumed that the systems
would become stressed by conditions far outside of those previously experienced. Further, we suggest that the larger
the number of future climate projections that exceed or fall far below their historic range, the more vulnerable a system
associated with a particular climate will be with respect to climate-induced changes. Our hydrologic landscape
vulnerability analysis (HLVA) places modern and projected environmental changes in the context of available historic
data. The HLVA assesses vulnerability to changes in temperature, precipitation, potential evapotranspiration, snow
accumulation, climatic moisture, surplus water, and seasonality of the water surplus by identifying areas that are
projected to experience deviations from historic conditions.
The ten future climate projections (for the 2041–2070 period) were compared to the decadal averaged data from 1901–
2010 for each AU. We calculated the historical standard deviation of each metric for each AU within the project area.
For each metric, we assume that any projection that is within two-standard deviations of the historical climate values
does not contribute to an increase in vulnerability, whereas projections outside of that range increase the vulnerability.
We then define vulnerability for a given index as the number of the ten projections that are outside of the historical
two-standard deviation threshold. Thus, the HLVA index assesses the likelihood that a given metric will exceed a two-
standard deviation threshold from the decadal mean under future climate scenarios.  A vulnerability index of ten
indicates that all ten climate projections were beyond two-standard deviations from the historical mean and so are
expected to experience projected conditions that they are not adapted to. The least vulnerable areas will have an index
of zero, which indicates that all future climate projections fell within the two-standard deviation threshold to which
systems are adapted to. The use of standard deviations is not an appropriate threshold metric for seasonality, because
it is a categorical variable. For the seasonality metric, any projected seasonality value that has not been observed
decadally between 1900 and 2010 increases the seasonality vulnerability index. For example, consider an AU that had
predominantly experienced Spring seasonality, with the occasional Fall seasonality and that 7 of 10 climate models
project Fall seasonality and 3 of 10 models predict Winter seasonality for 2041–2070. Since Winter seasonality was
not observed for any decade between 1900 and 2010, the three predictions for Winter seasonality each contribute to
the vulnerability index for seasonality. Finally, we analyzed the dominant HL code by area of the most vulnerable
AUs (those having a vulnerability index greater than seven on a scale of ten) for each metric in order to gain insight
about the dominant HL characteristics that relate to hydrologic vulnerability.
**2.5 Locational time series analyses**
Forty-five locations (Fig. 1 and Table 2) were selected for potential applications of the HL approach, based in part to
demonstrate the method's relevance to potential water resource stakeholders to identify areas where we thought results
could be of use to land managers. The time series for the decadal averages for each of the seven HL metrics were
analyzed for the AUs associated with each of these locations. Decadal averages were plotted at the decadal midpoint
for each 10-year period from 1901 to 2010. In addition, the 1971–2000 normal average for each variable and ten
climate projections (2041–2070) were plotted in a similar manner. The HLVA was then used to determine the mean
vulnerability index and the dominant HL code for the AUs associated with each location.





**3 Results**
**3.1 Hydrologic landscape summary**
Table 3 shows the percent coverage of the HL categories for the six states. Thirty percent of the region is mountainous
(elevation relief of AU > 300 m and < 10 % of AU area has slope < 1 %) and 7 % is flat (AUs with more than 50 %
area having < 1 % slope). The remaining area is classified as transitional. According to the soil permeability dataset
(Miller and White, 1998) produced from the STATSGO soils database (Soil Survey Staff, 2016), 98 % of the surface
soils (defined as the top 10 cm) are highly permeable (> 4.23 μm s$^{-1}$). Stratton et al. (2016) and Comeleo et al. (2014)
classified the subsurface permeability of the six-state region as 60 % high permeability and 40 % low permeability.
In terms of the 1971−2000 climate normal period, most of the area has the highest monthly water availability
(seasonality) during the winter (63 %), fall (24 %), spring (13 %), with approximately 1 % experiencing summer
seasonality. In addition, 30 % of the area is classified as having a moist, wet, or very wet climate, while 70 % is dry,
semi-arid or arid. The HL maps for the study area (Washington, Oregon, Idaho, California, Nevada, and Arizona) are
included in the appendix (Fig. A1). HL maps for the remainder of the conterminous US are also available and are also
included as supplemental material (Fig. S1). Note that the subsurface permeability maps were not extended across the
lower 48 states prior to submission but are available as supplemental material.
**3.2 Climate analyses**
**3.2.1 Regional (spatially continuous) time series analyses**
Figure 3 contains spatial trends in the change in FMI for the western U.S., showing wetter or drier decades relative to
the 1971−2000 baseline period (Figure S2 in the supplemental material illustrates similar data for the continental US).
Figure 4 displays projections of future (2041−2070) FMI values for the western U.S. relative to the 1971-2000 normal
period, based on the ten climate projections (Figure S3 in the supplemental material illustrates similar data for the
continental US).  Three of the climate models (CCSM-R4, MRI-CGCM3, and CESM1) indicate that portions of the
western U.S. may be wetter (as indicated by the blue areas in Fig. 4), while other areas will be drier (red) than or
similar to the 1971−2000 normal. Similarly, the maps suggest that seven of the climate models (CCSM4, GFDL,
inmcm4, CanESM2, HadGEM, CSIRO, and MIROC) project that much of the western U.S. will be considerably drier
than the normal period. The remaining models indicate that some areas will be slightly drier, whereas much of the
area will be similar to the 1971−2000 normal condition.
Figure 5 illustrates where the seasonal classes of surplus water have varied between 1901 and 2010 relative to the
1971−2000 base period (Figure S4 in the supplemental material illustrates similar data for the continental US). Most
areas throughout this historical period show little variation in the season of maximum available water (i.e., are shown
in white), but there are patterns in the water surplus seasonality that can be observed in the West. The 1940s, 1960s,
1980s, and 2000s seem to show later seasonality in southern Oregon and Idaho and Northern California and Nevada.
In contrast, portions of Oregon, Washington, and Arizona are shown to have earlier seasonality in the 1900s, 1910s,
1930s, 1950s, and 1970s.
Figure 6 illustrates the seasonal changes in surplus water as projected by the ten climate models for 2041−2070
compared to 1971−2000 (Figure S5 in the supplemental material illustrates similar data for the continental US). In
general, most of the climate models predict earlier surplus water in many of mountainous areas in the six western





states. Although most mountainous areas in Nevada are projected to have little change in seasonality, those that are
projected to change are projected to have earlier seasonality. In Arizona, the White Mountains are predicted to have a
later seasonality in two of ten climate projections (MIROC and GFDL), whereas seven projections predict earlier
seasonality in western Arizona.
**3.2.2 Vulnerability analyses**
The vulnerability maps (Fig. 7) identify areas that are more or less subject to extreme future climatic and hydrologic
variability (Similar vulnerability maps for the continental US are included in the supplemental materials (Fig. S6)).
All climate projections indicate that temperature will change almost ubiquitously across the Pacific west, however
changes in precipitation are much more spatially variable. The cold deserts and Mediterranean California Ecoregions
(Level 2) are more consistently projected to experience changes in precipitation than has been observed since 1901 on
a decadal basis. In contrast, major portions of Arizona, Washington, Oregon, and California have areas with low
vulnerability to change with respect to precipitation. The Hamon (1961) method of calculating monthly PET uses
temperature as the major input, so it is not surprising that the PET vulnerability map is similar to the temperature
vulnerability map. The April 1 snow accumulation (snow water equivalent) vulnerability map seems to indicate that
snow accumulation will change in many mountainous areas throughout the west, but particularly in the transitional
areas when compared to the most snow prone areas of the West. S' is a measure of available water (excess water
available for soil infiltration or overland flow). The map for S' suggests that the Warm Desert and Marine West Coast
Forest Ecoregions are more likely to experience substantial changes in available water (i.e., high vulnerability) in the
future. The FMI is calculated from the ratio of PET and precipitation per Eq. (1). The FMI vulnerability map indicates
that the Cold Desert Ecoregions of central, Western Washington, the Warm Deserts of Southern California, and High
Elevation Sierra Madre Mountains of south eastern Arizona are more likely to see substantial changes to the FMI. The
regional time series analyses (below) provide more information about whether those areas are expected to become
wetter or drier. The seasonality vulnerability map identifies AUs that are likely to have changes in seasonality. Portions
of the Sierra-Nevada Mountains in California and the Cascades in Oregon, and mountainous areas in Idaho are
projected to be more vulnerable to changes in seasonality. All other areas are not projected to be vulnerable to changes
for seasonality.
**3.2.3 Study area as a hydrologic landscape**
Table 4 summarizes an analysis of the HL classifications of the most vulnerable AUs for each metric. For example,
75 % of the AUs identified as vulnerable for snow accumulation were classified as dry, moist, or wet, therefore very
wet, semi-arid, and arid AUs are less likely to be vulnerable to changes in snow accumulation. Likewise, 76 % of AUs
vulnerable to changes in seasonality had a spring seasonality during the 1971−2000 normal period. The physical
properties represented by the dominant HL classes in Table 4 could help determine how various climate vulnerabilities
are ultimately expressed. For example, vulnerability to changes in snow or FMI mostly occur in regions with wetter
climates (Moist, Wet, or Very Wet climate), with fall or spring Seasonality, in areas with low subsurface permeability.
This could result in increased precipitation, with quicker runoff in areas that currently have delayed release of water.
Similarly, areas vulnerable to changes in surface runoff are arid landscapes with winter seasonality and highly





permeable subsurface parent materials. This means that these changes in runoff could have a large impact on
subsurface recharge and, ultimately, baseflow.
**3.2.4 Locational time series**
Historic and future changes in ecologically relevant variables are shown for three example locations (Napa-Sonoma
Valley, Willamette Valley, Mt. Hood; Fig. 8). Similar analyses have been performed for areas of ecological, economic,
or social significance (Table 2; see Appendix A (Fig. A2)). The number in the lower left corner of each graph in Fig.
8 indicates the vulnerability index for the specific metric and location. The vulnerability index for each location is
also listed in Table 2 for each metric. For instance, precipitation at Mt. Hood has a vulnerability index of '3', which
indicates that three of the climate projections exceed the threshold of two-standard deviations from the historic mean.
Table 2 indicates that 81 % of the 834 km$^2$ area analyzed for Mt. Hood (Site #7) had an HL code of VsHMH, (very
wet climate with spring seasonality, high subsurface permeability, mountainous terrain, and high surface
permeability). During the normal period, sixty-one percent of the 1867 km$^2$ Napa-Sonoma Valley (Site #26) had an
MwHMH HL classification, thus much of the area was classified as having a moist climate with winter seasonality,
high subsurface permeability, mountain terrain, and high surface permeability. Eighty-three percent of the 1234 km$^2$
Willamette Valley AUs (Site #8) had an HL code of WfHTH during the normal period. Overall, the Willamette Valley
had a wet climate, dominated by fall seasonality, high subsurface permeability, transitional terrain, and high surface
permeability.
The time series in Fig. 8 (and Fig. A2) illustrate the trend in average decadal temperature, precipitation, SWE, PET,
S', climate, and seasonality of water surplus. Note that each future (2041−2070) climate projection represents a single
data point that represents the 2041 − 2070 30-year range and is connected to the 2001−2010 decade with a dotted red
line. Additional figures for 41 other locations are provided in Appendix A (Fig. A2). Each of the three example areas
is predicted to be warmer in the 2041−2070 future climate projections. Further, these projected temperatures are almost
always outside of the historic (1901−2010) temperature range, and so all locations have high vulnerability with respect
to future temperatures. None of the three examples show a strong trend relating to future precipitation projections. Mt.
Hood appears to show increasing precipitation since 1901, but there is no evidence that the projected increases in
precipitation are outside of historic behavior. Napa-Sonoma and the Willamette Valley have low vulnerability for
change in snow, while Mt. Hood has high vulnerability for less April 1 snow accumulation in the 2041−2070 period.
PET is calculated directly from temperature and therefore shows trends strongly correlated to temperature. There are
no obvious trends in S' for the future projections for the selected examples; vulnerability of these sites for S' is low
to moderate. The FMI projections for Napa-Sonoma Valley, the Willamette Valley and Mt. Hood are outside of two-
standard deviations of historical trends in three to four out of ten of the projections (Table 2). In terms of seasonality,
the vulnerability index is equal to zero in the Willamette and Napa-Sonoma Valleys. For Mt. Hood, vulnerability is
low, with all of the future climate projections indicating that there will no longer be spring seasonality (the
predominant historical season for runoff), but only 3 projections suggest that seasonality would transition to a winter
seasonality that is not modeled to have occurred since at least 1900 on a decadal scale.



## 4 Discussion

Vulnerability maps (Fig. 7) were developed that indicate what areas across the landscape are projected to experience conditions that exceed two-standard deviations of the historic decadal average conditions. These maps provide spatially explicit details about the areas of the landscape that are most likely to experience conditions outside of those observed previously for seven different climate indicators. These maps were developed to facilitate long-term planning for stakeholders to be able to assess their risk to climatic impacts. It is possible that ecosystems, businesses, and communities in areas mapped as vulnerable may not be able to adapt to the stresses imposed by future environmental conditions.

From the vulnerability maps (Fig. 7), it is apparent that temperature [similar to Nijssen et al. (2001)] and PET are consistently projected to exceed the two-standard deviation threshold of historic conditions for most regions, though changes in PET may be overestimated (Johnson et al., 2012; U.S. Environmental Protection Agency, 2013). Precipitation vulnerability maps are not as spatially uniform as temperature. The vulnerability maps for snow accumulation and S' (surplus water available for runoff or infiltration) show that the areas mapped as most vulnerable for the two metrics are almost reversed, other than central Idaho and the coastal areas of California, Oregon, and Washington. According to the snow vulnerability map, it appears that most areas that receive much snow are projected to experience significant changes in future snow accumulation. In a related study on snow cover, Nolin and Daly (2006) found that the areas with the warmest winter temperatures are most at risk of having no snow cover in the future. Regarding the Feddema Moisture Index, Fig. 7 suggests that most of the models indicate that the magnitude of the FMI change is mostly within two-standard deviations of normal. The seasonality vulnerability map (Fig. 7) shows that the high Sierra-Nevada mountains in California, the Cascade mountains, and the mountainous areas in Idaho are somewhat prone to changes in seasonality.

We used a retrospective analysis of PRISM climatic time series data to gain an understanding of the distribution of environmental conditions present since 1901. While others have mapped resource and hydrologic vulnerability (Hill et al., 2014; Nolin and Daly, 2006; Vorosmarty et al., 2000; Winter, 2000), we are aware of few that have used retrospective analyses to inform the mapping efforts (Deviney et al., 2006; Kim et al., 2011; O'Brien et al., 2004) and are not aware of studies that have mapped resource vulnerability at a large scale using these types of data. It is important to emphasize that our definition of vulnerability is based on agreement of models with respect to climate conditions that are outside of historic ranges. The inference is that systems dependent on historic climate conditions may not be adapted to future conditions, and so are vulnerable. It is possible that they have the adaptive capacity to maintain their ecological and economic systems, but this is not a certainty. The vulnerability maps do not show, however, watersheds or communities downstream of these source areas that would be impacted by these changes.

For this analysis, the 30-year normal climate conditions are compared to decadal (10-year) climate conditions since 1901. In addition, the 30-year normal for future projections (2041-2070) is compared to the historic range of decadal climate data. While this may appear to be a discrepancy in the analysis, it was included intentionally to represent a conservative approach to quantifying vulnerability indices. Normal conditions are averaged over a 30-year period and therefore exhibit less variability than decadal averages or annual averages. By examining the past variability of the decadal averages since 1901, we use a period that exhibits variability without being an entirely smooth dataset. We





then compare that to the 30-year future climate normal, which inherently has much less variability. By using this
approach, we recognize that we are not treating past data in the same manner as we treat future climate projections.
We suggest that the resulting vulnerability conclusions are conservative, because if we had used decadal projections
for future climate data, the range of output would have been more variable. Decadal data would potentially have
increased our vulnerability indices for all parameters except those that are already at the maximum but should not
have decreased the index in any case.
In Fig. 8, examples are provided (Napa-Sonoma Valley, Willamette Valley, and Mt. Hood) to illustrate how analyses,
like the HLVA approach, can assist natural resource managers, business owners, or other stakeholders to understand
the potential impacts that changes in climate may have on their environment and the local bottom line. It is necessary
for a stakeholder to have an idea of the parameters most important to their ecosystem, industry, or resource of interest,
and it should prove useful for land and resource managers that are seeking location specific information about potential
climatic impacts (Glick et al., 2011; Lawler et al., 2010).
Important stakeholders in the western U.S. that may be expected to experience impacts from hydrological changes
associated with climate include the wine and skiing industries. The Napa-Sonoma and Willamette Valleys are
economically important for their grape vineyards and associated wineries. The Willamette Valley is recognized for
the quality of its pinot noir varietals (http://wine.appellationamerica.com/wine-region/Willamette-Valley.html), which
require narrower temperature ranges than other grape cultivars (Burakowski and Magnusson, 2012; Jones et al., 2010).
Due to the importance of the pinot noir varietal to viticulturists in the Willamette Valley, they are likely more
concerned with changes in temperature than FMI. The Napa-Sonoma region is recognized for a wider variety of grape
cultivars (http://wine.appellationamerica.com/wine-region/Napa-Valley.html, Elliott-Fisk, 1993) that have higher
tolerance for temperature fluctuations than the pinot noir varietals commonly grown in the Willamette Valley (Jones
et al., 2010). Figure 8 indicates that both the Willamette Valley and Napa-Sonoma have temperature vulnerability
indices of ten out of ten, and both have FMI vulnerability indices of three out of ten. These index values suggest that
both locations are projected to have future temperatures that are significantly different than the historic observed
temperatures. However, the Willamette Valley pinot noir vineyards may have more cause for concern, since pinot noir
grapes are documented to be more sensitive to temperature. In the Napa and Sonoma Valleys, there may be less need
for concern with temperature than in the Willamette Valley. In addition, while both locations have the same FMI
vulnerability indices, Fig. 8 illustrates that FMI projections for Napa-Sonoma are much more variable than for the
Willamette Valley. Thus, there is more uncertainty in the modeled water availability for Napa-Sonoma. Taken at face
value, these modeled results suggest that a vintner growing warm temperature grape species in the Willamette Valley
may have more confidence in his investments relative to a vintner in Napa-Sonoma, where there is more uncertainty
regarding long-term water availability.
The skiing industry is also an important economic contributor. According to Burakowski and Magnusson (2012), the
difference in economic impact between a high and low snowfall year for the State of Oregon is $38.1 million, while
California is estimated to lose more than $75 million in low snow years. Mt. Hood is well known for its recreational
snow sports and winter tourism in Oregon and would be impacted differently by the seven metrics than the Willamette
and Napa-Sonoma examples (Fig. 8). Thus, resource managers and business leaders at Mt. Hood are likely more





concerned about snow accumulation in their watershed than those in the wine and grape industries (although grape
grower's ability to irrigate may be impacted by snow accumulation in the region). According to our analyses, Mt.
Hood has a snow vulnerability index of seven out of a maximum of ten. The analysis of seasonality suggests some
chance of a shorter ski season due to the spring runoff occurring earlier during the winter season. Even though these
conditions have occurred in the past (Fig. 8), this may be much more deleterious to the economics of the modern or
future ski industry than it was in the 1900s, because it contributed much less to the historic economy.
The quantity (as indicated by the FMI) and timing (as indicated by the seasonality of the water surplus) of moisture
availability only account for a portion of the water balance for an area. The FMI and seasonality are assumed to be
proxies for the quantity and timing of moisture availability, but when moisture is available as surface runoff, it may
then infiltrate into the ground or act as surface runoff. Water may infiltrate the surface layer of soil (depending on the
soil permeability) and may enter the subsurface layers (depending on the vertical conductivity of the subsurface
layers). The velocity of water through the subsurface layers that flows towards a stream channel depend upon the
horizontal conductivity of the subsurface layers. Thus, if the water was retained as surface or subsurface runoff, it may
be transported more quickly in the downhill direction and into a stream channel depending upon the steepness of the
terrain (included in the HL classification). As it relates to streamflow, the unique combination of the five HL
characteristics (climate, seasonality, surface permeability, subsurface permeability, and terrain) allows for the
estimation of catchment hydrologic responses to changes in temperature and climate (Leibowitz et al., 2014; Patil et
al., 2014). The HL approach has proved useful for streamflow prediction in gaged basins for some HL classes and
should be useful in many ungaged basins as well. However, this paper illustrates how the HL approach can help to
assess climatic and hydrologic vulnerability across large spatial scales. The three examples we provided, show how
the HLVA method could be useful to resource managers for considering how future climate conditions may impact
important economic and conservation resources (for additional examples refer to the appendix (2).

## 5 Summary and conclusions

The hydrologic landscapes (HL) concept has proved useful for gaining a better understanding of hydrologic behaviour
at the assessment unit and watershed scales across large geographic regions. By applying the HL concept to climatic
and vulnerability analyses, we provide a planning approach that allows resource managers to consider historic and
projected climate behavior in their long-term planning efforts so they can better assess the risk imposed by potential
changes.  The methodology also allows stakeholders to focus on particular areas of interest, which provides the
flexibility necessary for the information to be relevant across applications and sectors. By applying the modified
Wigington et al. (2013) HL approach across the western US, resource managers will gain a better understanding of
the projected vulnerability of water resource availability in a large portion of the United States.

## 6 Data availability

The geospatial data files (Jones et al., 2020) will be uploaded to the GeoPlatform (https://www.geoplatform.gov) and
EPA Environmental Dataset Gateway (https://edg.epa.gov). Data cannot be made publicly available and the DOI link
cannot go activated until the paper is published per internal US EPA policy.




**7 Code availability**

Authors may deposit code in a FAIR-aligned repository/archive upon final acceptance of the manuscript for publication.

**8 Video abstract**

No video abstract is available at this time.

**9 Author contribution**

CJ and SL conceptualized the study with significant input from KS. CJ performed the formal analyses, investigation, developed the methodologies (with input from SL, KS, and RC), managed the project, developed the model code, performed the analyses, developed the final figures and tables, and wrote draft versions of the manuscript, and incorporated co-author feedback into the final version of the manuscript. SL supervised the project and performed project administration. RC contributed technical expertise regarding spatial data analyses and familiarity with hydrologic landscapes data analyses. RC and LS developed the subsurface permeability datasets. PM and CW provided and advice regarding the use of the future climate projections and the processing of those datasets.

**10 Acknowledgements**

We would like to thank James Markwiese, Mohammad Safeeq, and Eric Sproles for their constructive feedback on the manuscript. We also appreciate Jim Wigington's insight and input on early drafts of our mapping products. We acknowledge the World Climate Research Programme's Working Group on Coupled Modelling, which is responsible for CMIP, and we thank the climate modeling groups (listed in Table 1 of this paper) for producing and making available their model output. For CMIP the U.S. Department of Energy's Program for Climate Model Diagnosis and Intercomparison provides coordinating support and led development of software infrastructure in partnership with the Global Organization for Earth System Science Portals. The information in this document has been funded entirely by the U.S. Environmental Protection Agency, in part through an appointment to the Internship/Research Participation Program at the Office of Research and Development, U.S. Environmental Protection Agency, administered by the Oak Ridge Institute for Science and Education through an interagency agreement between the U.S. Department of Energy and EPA, and also through Student Services Contract #EP-15-W-000041. The views expressed in this paper are those of the authors and do not necessarily reflect the views or policies of the U.S. Environmental Protection Agency. Any use of trade, firm, or product names is for descriptive purposes only and does not imply endorsement by the U.S. Government.

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





**12 Figures**

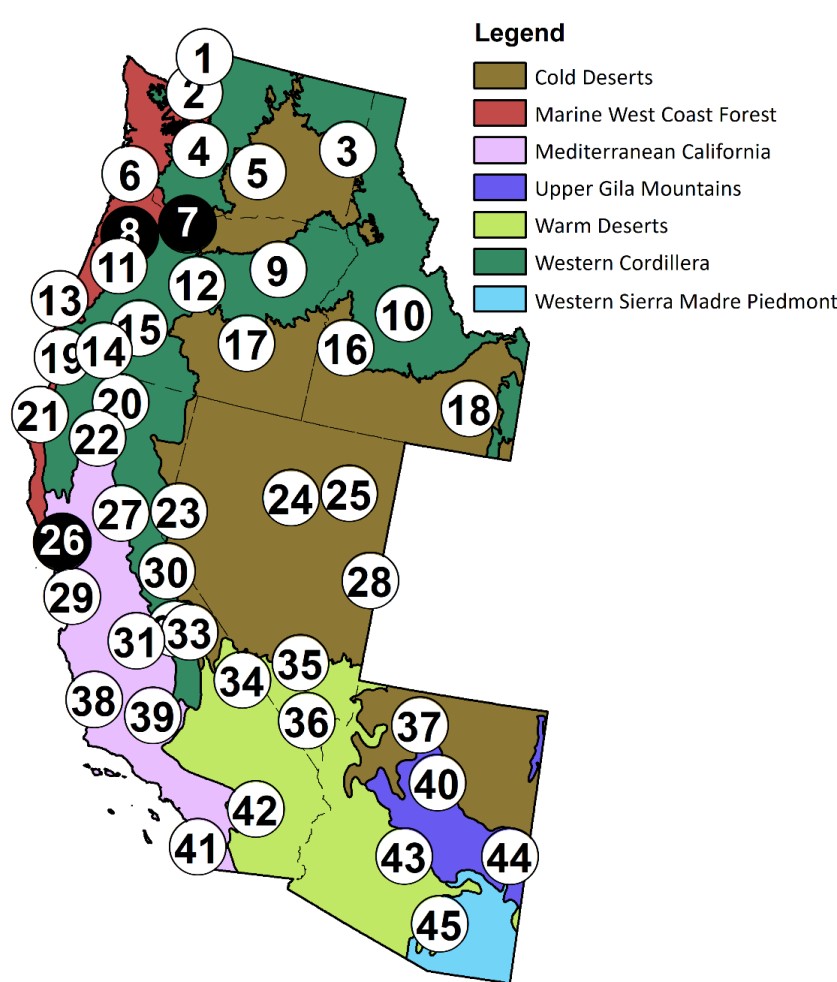


**Figure 1. Study area showing map with the six states of WA, OR, ID, CA, NV, and AZ. Also shown are the 7 EPA Level II**
**Ecoregions and 45 locations identified by numbered circles with three example locations in black circles (Table 2).**



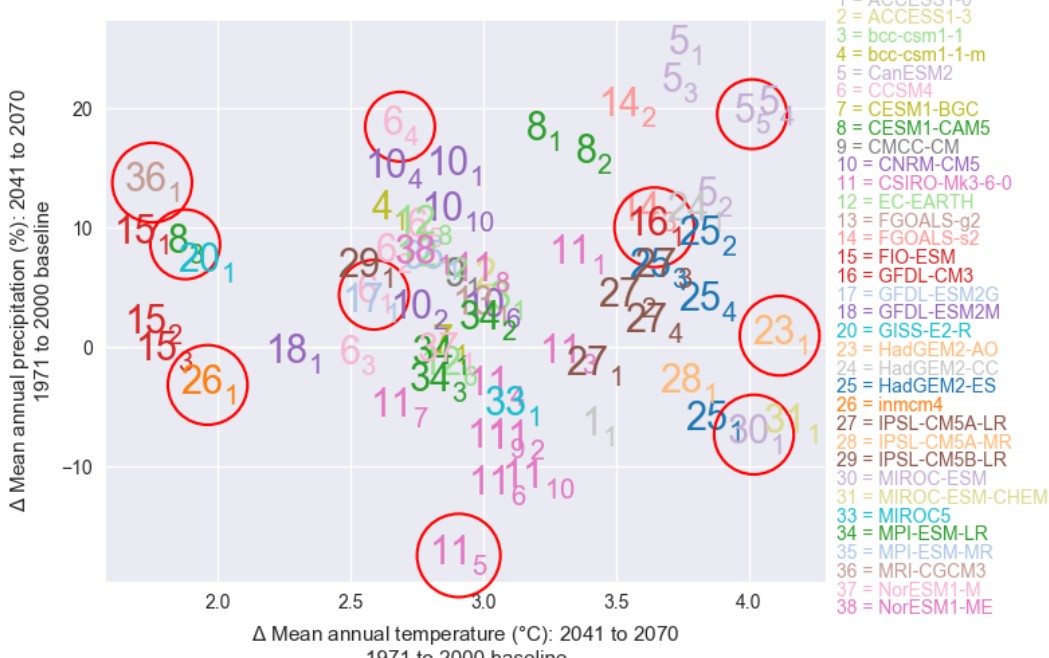

**Figure 2. Scatterplot showing the range of mean temperature and precipitation projections for the 2041−2070 climate models across the study area. The circled data points identify the climate projections used in our analyses.**


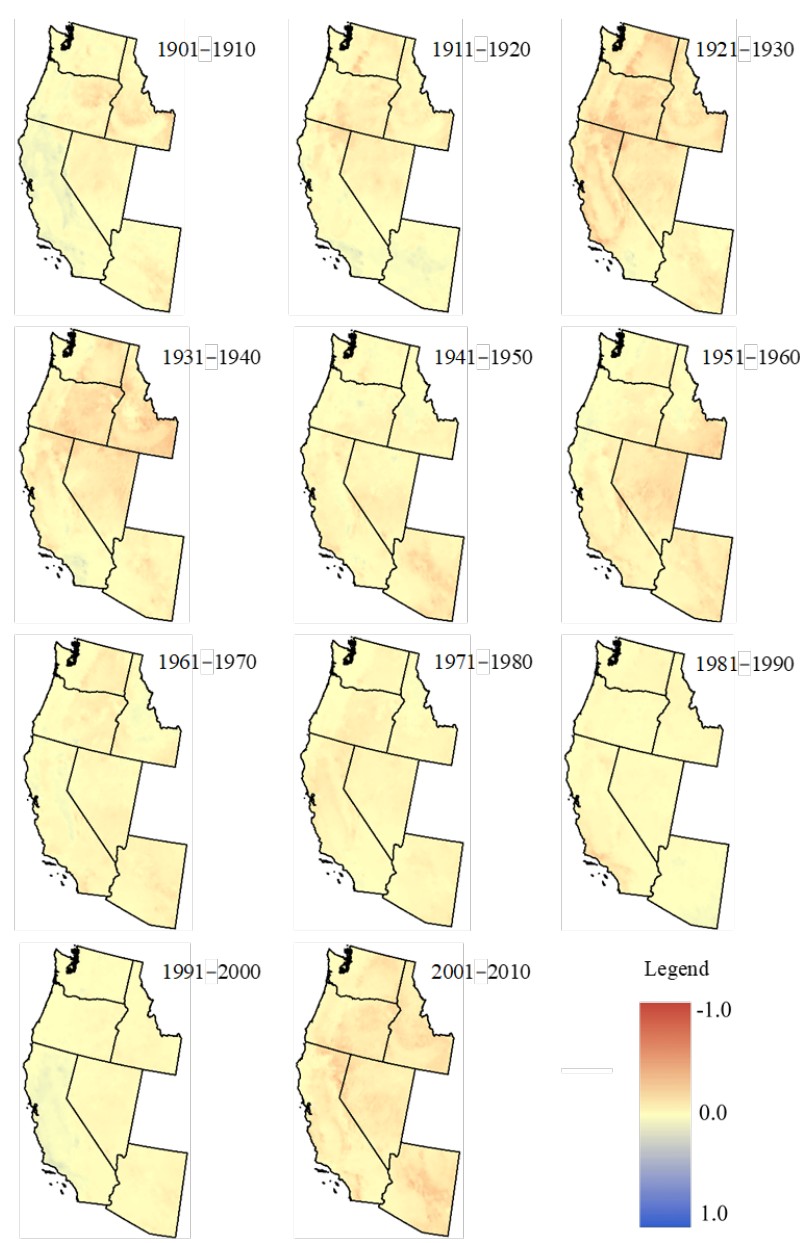


**Figure 3. Decadal change in Feddema Moisture Index relative to 1971−2000 normal period. Red and blue colors indicate drier and wetter average conditions than 1971−2000, respectively.**

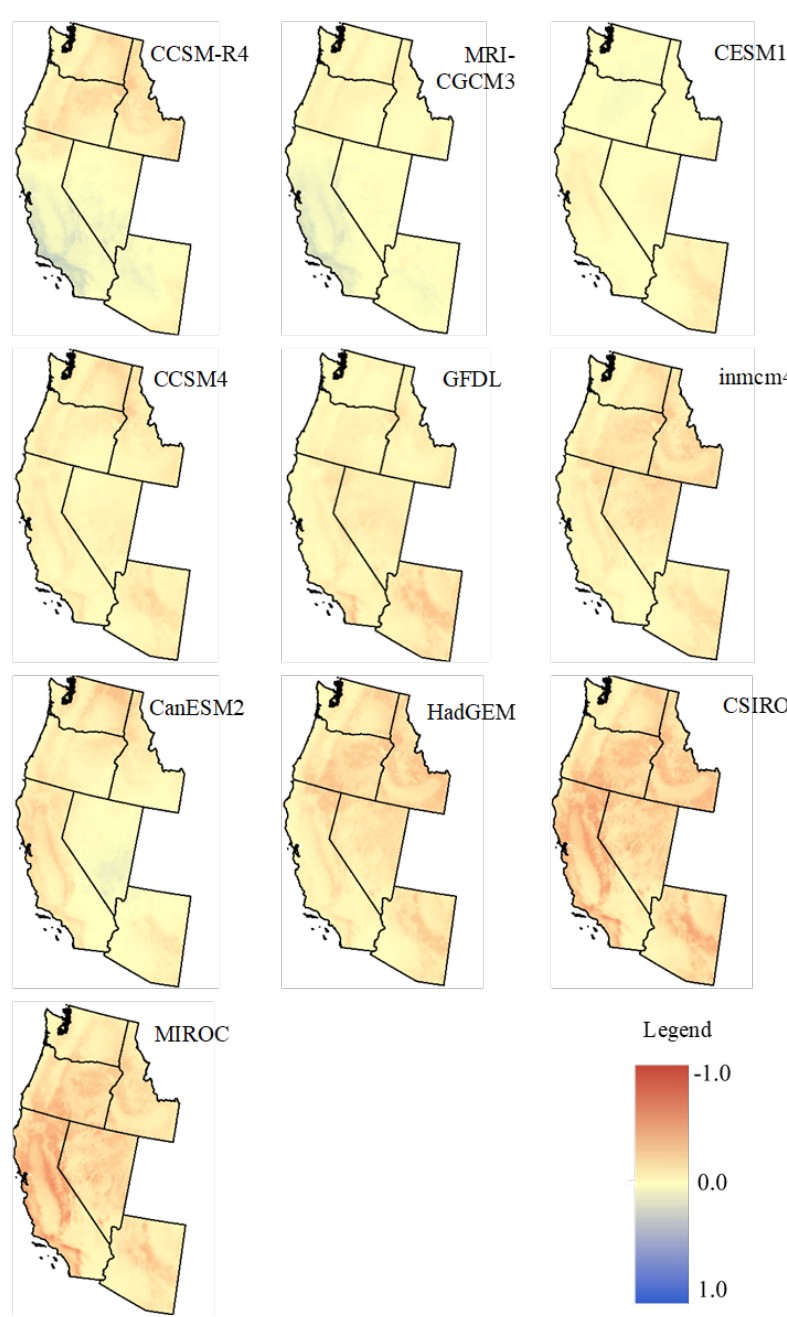


**Figure 4. Projected change in Feddema Moisture Index for 2041−2070 relative to 1971−2000 for ten climate models (Table 1). Red and blue colors indicate drier and wetter conditions than the 1971−2000 base period, respectively. Abbreviated model names correlate to those in Table 1.**





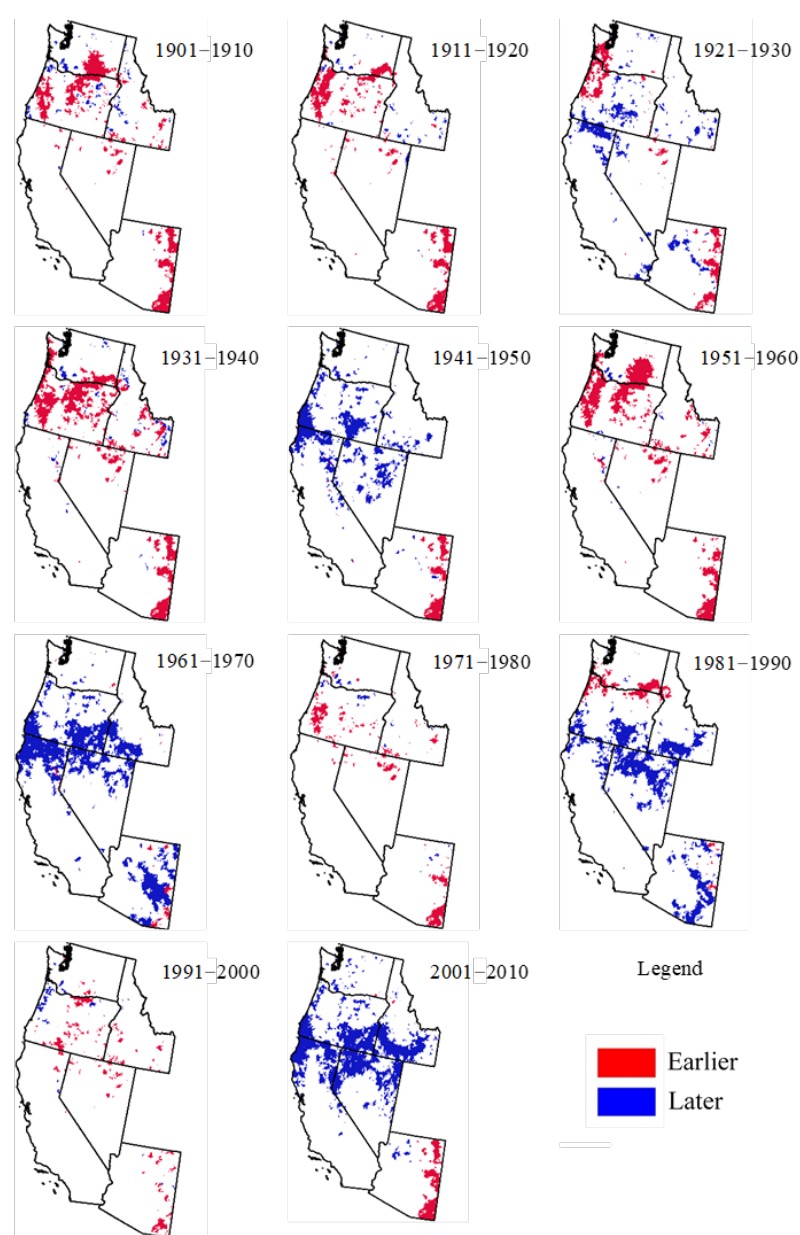


**Figure 5. Decadal change in seasonality of water surplus since 1901 relative to 1971−2000. Red and blue colors indicate earlier and later seasonality than the 1971−2000 base period, respectively.**



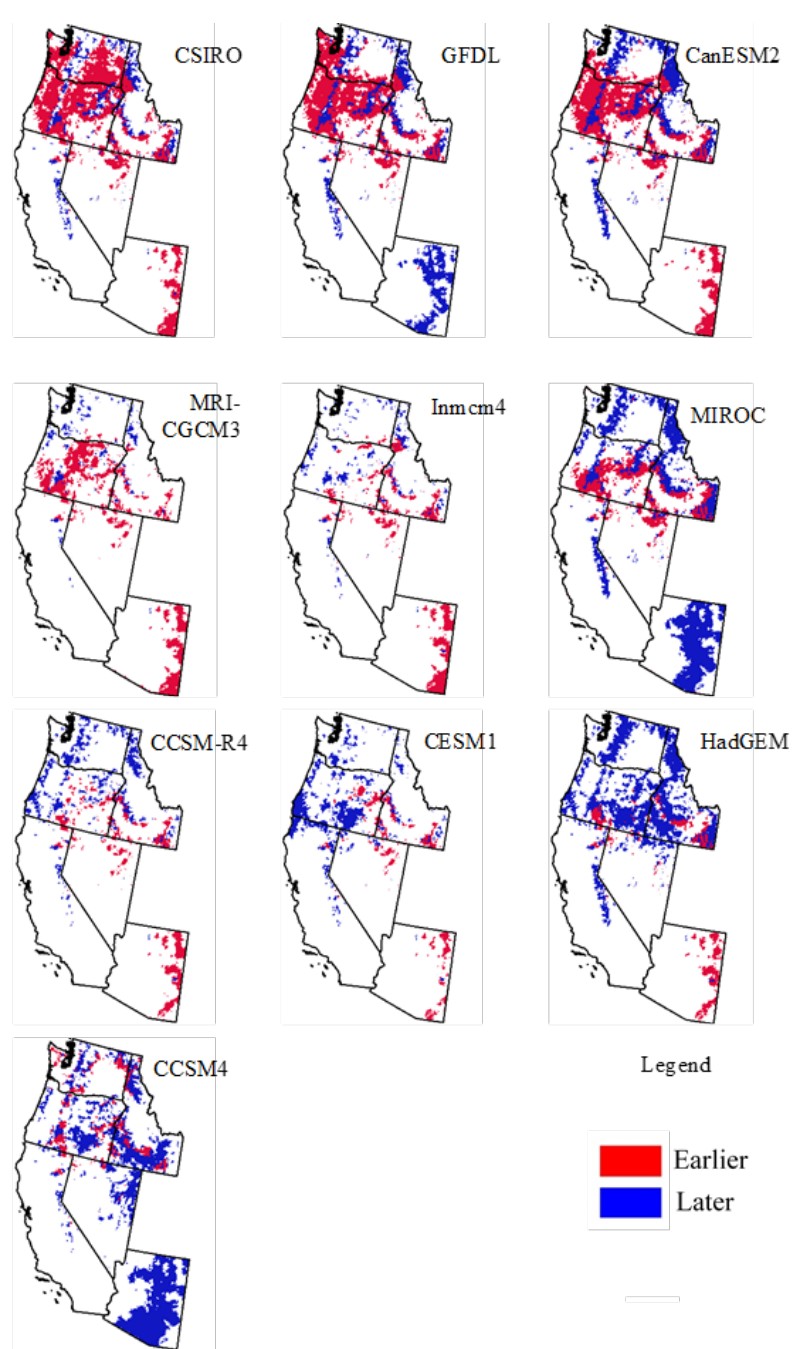


**Figure 6. Projected change in seasonality of water surplus for 2041−2070 relative to 1971−2000 for ten climate models. Red and blue colors indicate earlier and later seasonality than the 1971−2000 base period, respectively. Abbreviated model names correlate to those in Table 1.**



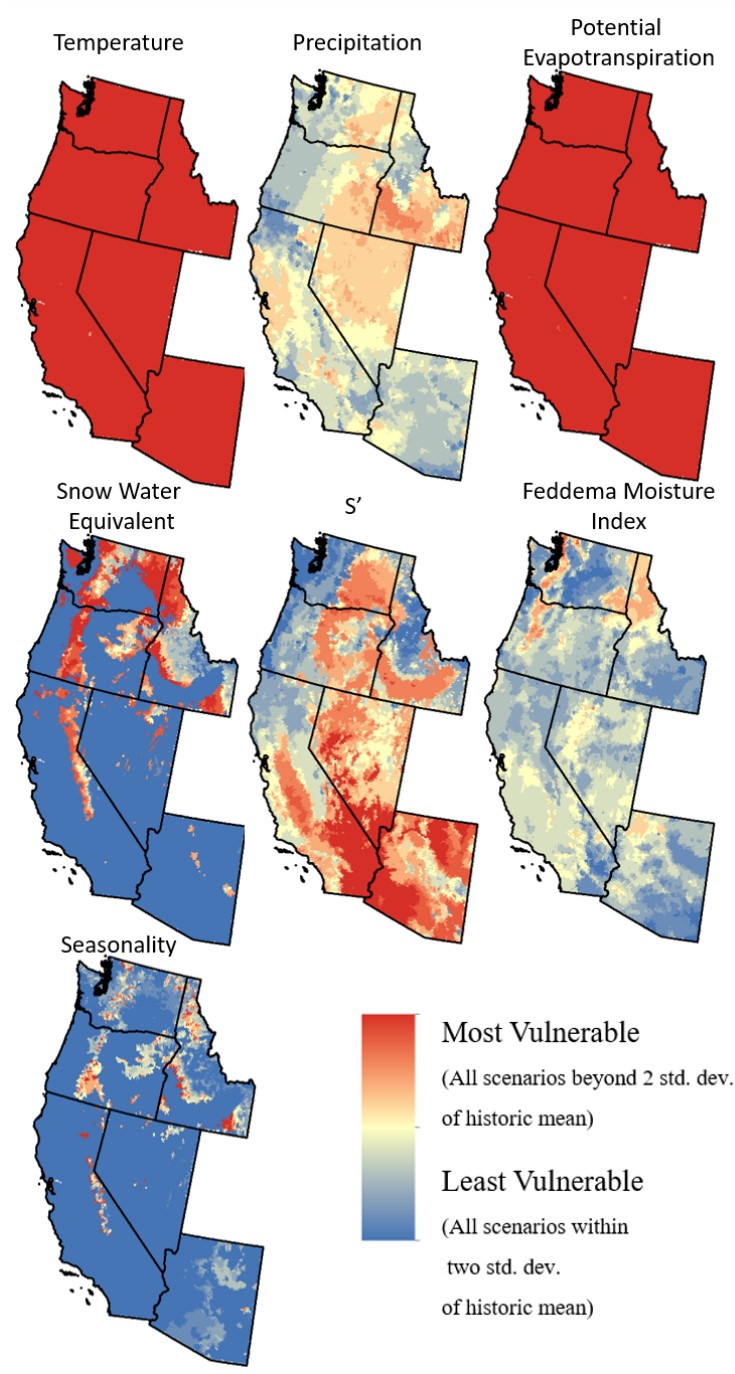


**Figure 7. Vulnerability indices for temperature, precipitation, potential evapotranspiration, snow water equivalent (April 1), S' (available water), Feddema Moisture Index, and seasonality. The least vulnerable locations are those projected to be within two-standard deviations of the historic (1901−2010) mean in all nine climate models.**

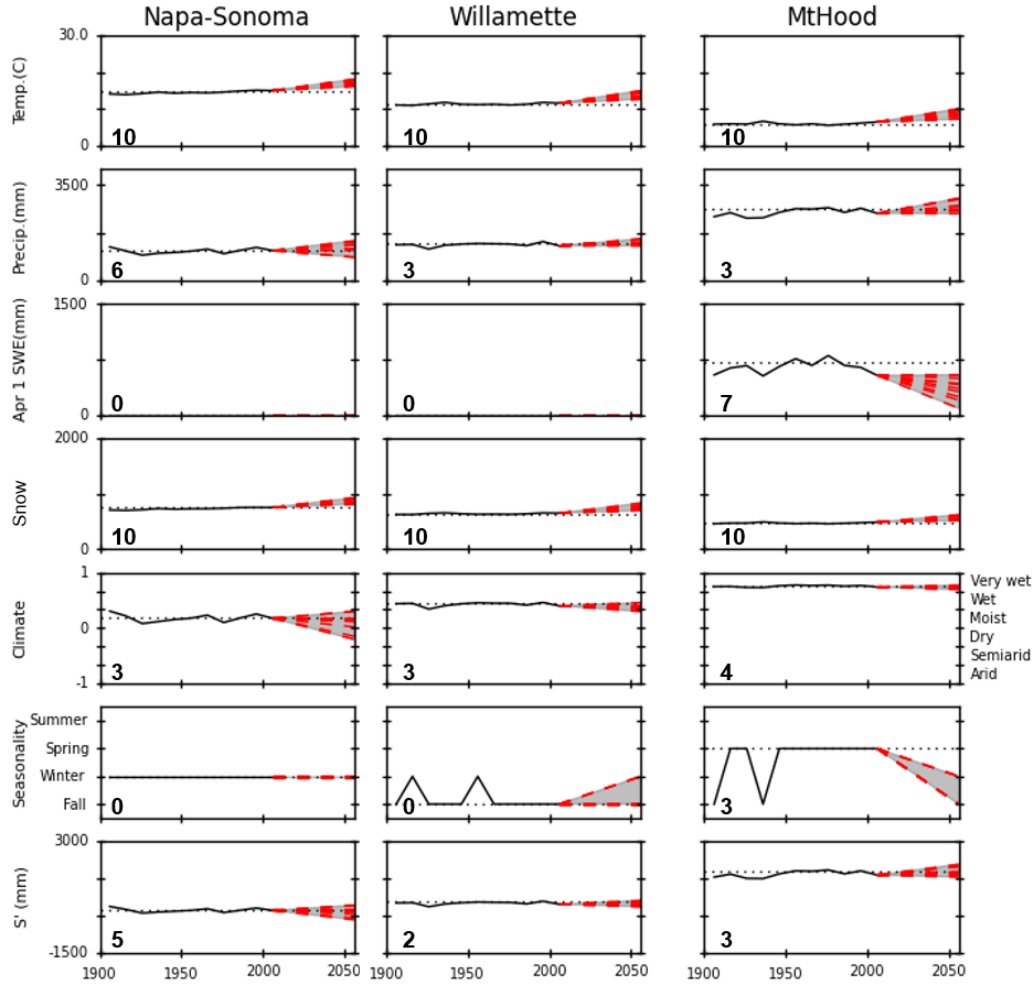


**Figure 8. Time series of average decadal temperature, precipitation, snow (April 1 snow water equivalent (mm)), potential evapotranspiration (PET), available water (S'), FMI, and seasonality for three specific locations in the western U.S. Dotted black line represents the 1971−2000 base period; the dashed red line connects the 2001−2010 value to the 2041−2070 climate projections. The number in lower left indicates the vulnerability index for the metric and location depicted in the associated graph.**





**13 Tables**

**Table 1. CMIP5 Climate model summary for 2041−2070 precipitation and temperature data (Bureau of Reclamation, 2014).**

| WCRP CMIP5 Climate Model | Model abbreviated name | Model realization used herein | Abbreviated name used in this paper for realization |
|---|---|---|---|
| Canadian Earth System Model | CanESM2 | r5i1p1 | CanESM2 |
| Community Climate System Model | CCSM4 | r1i1p1 | CCSM4 |
| Community Climate System Model | CCSM4 | r4i1p1 | CCSM4-R4 |
| Community Earth System Model | CESM1 | r3i1p1 | CESM1 |
| Commonwealth Scientific and Industrial Research Organisation Mark 3.6 | CSIRO-Mk3-6-0 | r5i1p1 | CSIRO |
| Geophysical Fluid Dynamics Laboratory Coupled Climate Model | GFDL-CM3 | r1i1p1 | GFDL |
| Hadley Global Environment Model | HadGEM2-AO | r1i1p1 | HadGem |
| Institute for Numerical Mathematics Climate Model | INM-CM4 | r1i1p1 | inmcm4 |
| Model for Interdisciplinary Research on Climate | MIROC-ESM | r1i1p1 | MIROC |
| Meteorological Research Institute | MRI-CGCM3 | r1i1p1 | MRI-CGCM3 |





781
782
783

**Table 2. Summary table for 45 study locations (sorted by decreasing latitude) provides numeric ID from Fig. 1, total analysis area, dominant HL class (representing climate, seasonality, subsurface permeability, terrain, and surface permeability), percent area represented by dominant HL class, latitude and longitude of the center point of the area, and vulnerability indices for temperature, potential evapotranspiration (PET), precipitation, S', snow, Feddema Moisture Index (FMI), and seasonality.**

| Site # | Name | Area (km²) | Dominant HL Class* | % Dominant Area | Coordinates Lat. | Coordinates Long. | Temp. | PE T | Precip. | S' | Snow | FMI | Seasonality |
|---|---|---|---|---|---|---|---|---|---|---|---|---|---|
| 1 | Bellingham | 212 | WfLTH | 99 % | 48.77 | -122.45 | 10 | 10 | 5 | 1 | 0 | 9 | 0 |
| 2 | Spokane | 592 | DfHTH | 80 % | 47.64 | -117.43 | 10 | 10 | 6 | 7 | 10 | 3 | 1 |
| 3 | Seattle | 669 | WfLTH | 78 % | 47.60 | -122.25 | 10 | 10 | 4 | 1 | 0 | 5 | 2 |
| 4 | MtRainier | 718 | VsLMH | 76 % | 46.85 | -121.79 | 10 | 10 | 4 | 2 | 7 | 4 | 2 |
| 5 | Yakima | 438 | SfHTH | 86 % | 46.63 | -120.60 | 10 | 10 | 3 | 6 | 0 | 0 | 0 |
| 6 | Portland | 932 | WfHTH | 67 % | 45.53 | -122.66 | 10 | 10 | 3 | 2 | 0 | 6 | 0 |
| 7 | MtHood | 834 | VsHMH | 81 % | 45.37 | -121.70 | 10 | 10 | 3 | 3 | 7 | 4 | 3 |
| 8 | UmatillaNF | 2,147 | MsLMH | 29 % | 44.87 | -118.70 | 10 | 10 | 6 | 3 | 6 | 3 | 4 |
| 9 | Willamette | 1,234 | WfHTH | 83 % | 44.84 | -123.14 | 10 | 10 | 3 | 2 | 0 | 4 | 0 |
| 10 | ChallisNF | 4,348 | WsLMH | 74 % | 44.55 | -114.75 | 10 | 10 | 6 | 0 | 3 | 2 | 0 |
| 11 | Bend | 948 | SfHTH | 68 % | 44.21 | -121.26 | 10 | 10 | 4 | 8 | 0 | 3 | 0 |
| 12 | Eugene | 523 | WfHFH | 64 % | 44.10 | -123.15 | 10 | 10 | 3 | 1 | 0 | 2 | 0 |
| 13 | Boise | 594 | SwHTH | 51 % | 43.61 | -116.24 | 10 | 10 | 8 | 8 | 0 | 2 | 0 |
| 14 | MalheurNWR | 1,355 | SwHFH | 69 % | 43.27 | -119.04 | 10 | 10 | 6 | 7 | 0 | 2 | 0 |
| 15 | CraterLake | 1,721 | WsHTH | 45 % | 42.98 | -122.08 | 10 | 10 | 3 | 2 | 9 | 3 | 10 |
| 16 | Pocatello | 349 | DwHTH | 45 % | 42.88 | -112.43 | 10 | 10 | 7 | 7 | 0 | 1 | 0 |
| 17 | SiskiyouNF | 926 | VwLMH | 100 % | 42.36 | -124.29 | 10 | 10 | 2 | 0 | 0 | 2 | 0 |
| 18 | Medford | 375 | DfLTH | 60 % | 42.34 | -122.89 | 10 | 10 | 1 | 5 | 0 | 2 | 0 |
| 19 | SixRivers | 1,527 | VwLMH | 100 % | 41.63 | -123.79 | 10 | 10 | 2 | 2 | 0 | 4 | 0 |
| 20 | MtShasta | 956 | WwHMH | 49 % | 41.36 | -122.23 | 10 | 10 | 1 | 2 | 0 | 3 | 0 |
| 21 | RubyMtn | 1,132 | DfLTH | 44 % | 40.68 | -115.31 | 10 | 10 | 6 | 5 | 9 | 4 | 0 |
| 22 | Arcata-HumboldtCo | 2,511 | WwLMH | 63 % | 40.62 | -124.01 | 10 | 10 | 3 | 2 | 0 | 3 | 0 |
| 23 | Redding | 478 | MwHTH | 59 % | 40.56 | -122.38 | 10 | 10 | 2 | 2 | 0 | 2 | 0 |
| 24 | BattleMtn | 902 | SwLMH | 75 % | 40.09 | -116.71 | 10 | 10 | 6 | 7 | 0 | 4 | 0 |
| 25 | Reno | 382 | SwHTH | 40 % | 39.54 | -119.80 | 10 | 10 | 4 | 7 | 0 | 3 | 0 |





| Site # | Name | Area (km²) | Dominant HL Class* | % Dominant Area | Coordinates | | Vulnerability Index | | | | | | |
|---|---|---|---|---|---|---|---|---|---|---|---|---|---|
| | | | | | Lat. | Long. | Temp. | PE T | Precip. | S' | Snow | FMI | Seasonality |
| 26 | GreatBasinNP | 38 | MsLMH | 100 % | 39.01 | -114.26 | 10 | 10 | 4 | 5 | 0 | 4 | 1 |
| 27 | Sacramento | 855 | SwHFH | 88 % | 38.57 | -121.39 | 10 | 10 | 6 | 7 | 0 | 3 | 0 |
| 28 | Napa-Sonoma | 1,867 | MwHTH | 61 % | 38.37 | -122.53 | 10 | 10 | 6 | 5 | 0 | 3 | 0 |
| 29 | YosemiteNP | 2,455 | VsLMH | 44 % | 37.93 | -119.55 | 10 | 10 | 4 | 4 | 9 | 3 | 0 |
| 30 | SanFranciscoBay | 3,356 | DwHMH | 19 % | 37.44 | -122.29 | 10 | 10 | 6 | 5 | 0 | 5 | 0 |
| 31 | SierraNF | 5,349 | WwLMH | 31 % | 37.17 | -119.05 | 10 | 10 | 4 | 4 | 0 | 2 | 0 |
| 32 | HighSierras | 2,239 | WsLMH | 32 % | 37.15 | -118.81 | 10 | 10 | 2 | 4 | 1 | 2 | 0 |
| 33 | NevadaTestSite | 3,121 | AwHMH | 67 % | 36.96 | -116.22 | 10 | 10 | 5 | 10 | 0 | 4 | 0 |
| 34 | Fresno | 1,393 | AwHFH | 100 % | 36.74 | -119.91 | 10 | 10 | 5 | 8 | 0 | 4 | 0 |
| 35 | DeathValleyNP | 7,862 | AwHMH | 50 % | 36.45 | -117.03 | 10 | 10 | 5 | 10 | 0 | 5 | 0 |
| 36 | LasVegas | 977 | AwHTH | 65 % | 36.23 | -115.26 | 10 | 10 | 4 | 10 | 0 | 4 | 0 |
| 37 | GrandCanyonNP | 3,475 | SwHMH | 28 % | 36.22 | -112.11 | 10 | 10 | 4 | 10 | 0 | 6 | 0 |
| 38 | SanLuisObispo | 2,653 | DwLMH | 98 % | 35.36 | -120.63 | 10 | 10 | 4 | 4 | 0 | 4 | 0 |
| 39 | Bakersfield | 3,399 | AwHFH | 96 % | 35.33 | -119.14 | 10 | 10 | 4 | 9 | 0 | 4 | 0 |
| 40 | Flagstaff | 365 | DwHMH | 51 % | 35.19 | -111.60 | 10 | 10 | 3 | 4 | 0 | 4 | 0 |
| 41 | JoshuaTreeNP | 2,599 | AwLMH | 68 % | 33.92 | -115.99 | 10 | 10 | 5 | 7 | 0 | 5 | 0 |
| 42 | WhiteMtns | 4,855 | WfLMH | 23 % | 33.87 | -109.53 | 10 | 10 | 4 | 3 | 0 | 3 | 0 |
| 43 | Phoenix | 2,304 | AwHFH | 63 % | 33.52 | -112.11 | 10 | 10 | 3 | 10 | 0 | 2 | 1 |
| 44 | SanDiego | 1,276 | SwLMH | 37 % | 32.90 | -117.06 | 10 | 10 | 4 | 6 | 0 | 4 | 0 |
| 45 | Tucson | 1,838 | AwHTH | 62 % | 32.19 | -110.95 | 10 | 10 | 3 | 9 | 0 | 1 | 2 |

*Climate class (1st letter): V=very wet; W=wet; M=moist; D=dry; S=semiarid; A=arid
Seasonality class (2nd letter): f=fall; w= winter; s=spring; u=summer
Subsurface permeability class (3rd letter): L=low; H=high
Terrain class (4th letter): M=mountain; T=transitional; F=flat
Surface permeability class (5th letter): L=low; H=high



**Table 3. Percent of area of each HL category and classification within the six-state region (1971−2000)**

| Category | Classification | Area (%) |
|---|---|---|
| Climate | Arid | 21 % |
| | Semi-arid | 34 % |
| | Dry | 15 % |
| | Moist | 9 % |
| | Wet | 14 % |
| | Very wet | 7 % |
| Season | Spring (AMJ[1]) | 13 % |
| | Summer (JAS[2]) | 1 % |
| | Fall (OND[3]) | 24 % |
| | Winter (JFM[4]) | 63 % |
| Subsurface Perm. | Low | 40 % |
| | High | 60 % |
| Terrain | Flat | 7 % |
| | Transitional | 63 % |
| | Mountain | 30 % |
| Surface Perm. | Low | 2 % |
| | High | 98 % |

[1]AMJ: April, May, and June
[2]JAS: July, August, and September
[3]OND: October, November, and December
[4]JFM: January, February, and March





**Table 4. Hydrologic landscape characteristics of assessment units identified as vulnerable (having a vulnerability index**
**greater than 7 on a scale of 10) for each metric.**

|  |  | % Assessment units that share HL classification | | | | | | | | |
|---|---|---|---|---|---|---|---|---|---|---|
|  |  | **Climate**[1] | | **Seasonality**[2] | | **Subsurface Perm.**[3] | | **Terrain**[4] | | **Surface perm.**[3] | |
| Vulnerability Parameter | Snow | 75 % | D, M, or W | 87 % | f or s | 53 % | L | 82 % | M | 100 % | H |
|  | FMI | 71 % | V or W | 65 % | f | 75 % | L | 75 % | M | 100 % | H |
|  | Seasonality | 75 % | W or M | 76 % | s | 51 % | H | 83 % | M | 99 % | H |
|  | S' | 92 % | A or S | 79 % | w | 75 % | H | 87 % | M or T | 99 % | H |
|  | ppt | 72 % | D or S | 79 % | f or w | 71 % | H | 97 % | M or T | 98 % | H |
|  | tmean | 70 % | D, S, or A | 87 % | f or w | 60 % | H | 93 % | M or T | 98 % | H |
|  | PET | 70 % | D, S, or A | 87 % | f or w | 60 % | H | 93 % | M or T | 98 % | H |

[1]A=arid, S=semiarid, D=dry, M=moist, W=wet
[2]f=fall, w=winter, s=spring
[3]L=low, H=high
[4]T=transitional, M=mountainous



**Appendix A**

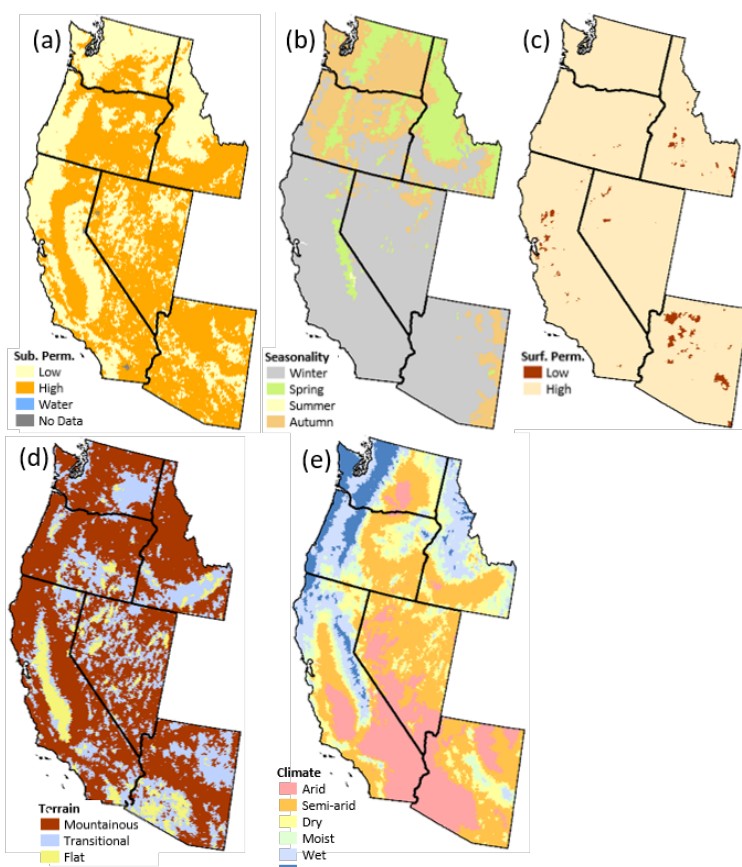


**Figure A1. Hydrologic Landscape maps of Washington, Idaho, Oregon, California, Nevada, and Arizona were used in the**
**HLVA analysis [(a) Subsurface Permeability, (b) Seasonality of precipitation surplus, (c). Surface permeability, (d) Climate,**
**and (e) Terrain]. Notes: The seasonality map for the PNW has been updated from the original Leibowitz 2016 HL map, as**
**we separated their winter seasonality into two seasons (winter and fall).**




**Figure A2 (Plates 1–15)**
**Time series of average decadal temperature, precipitation, snow (April 1 snow water equivalent (mm), potential**
**evapotranspiration (PET), available water (S'), FMI, and seasonality for specific locations identified in Fig. 1 and Table 2**
**in the western United States Dotted black line represents the 1971–2000 base period; the dashed red line connects the 2001–**
**2010 value to the 2041–2070 climate projections. Note that Oregon, Washington, and Idaho locations are displayed first in**
**alphabetical order and are followed by those of California, Nevada, and Arizona.**

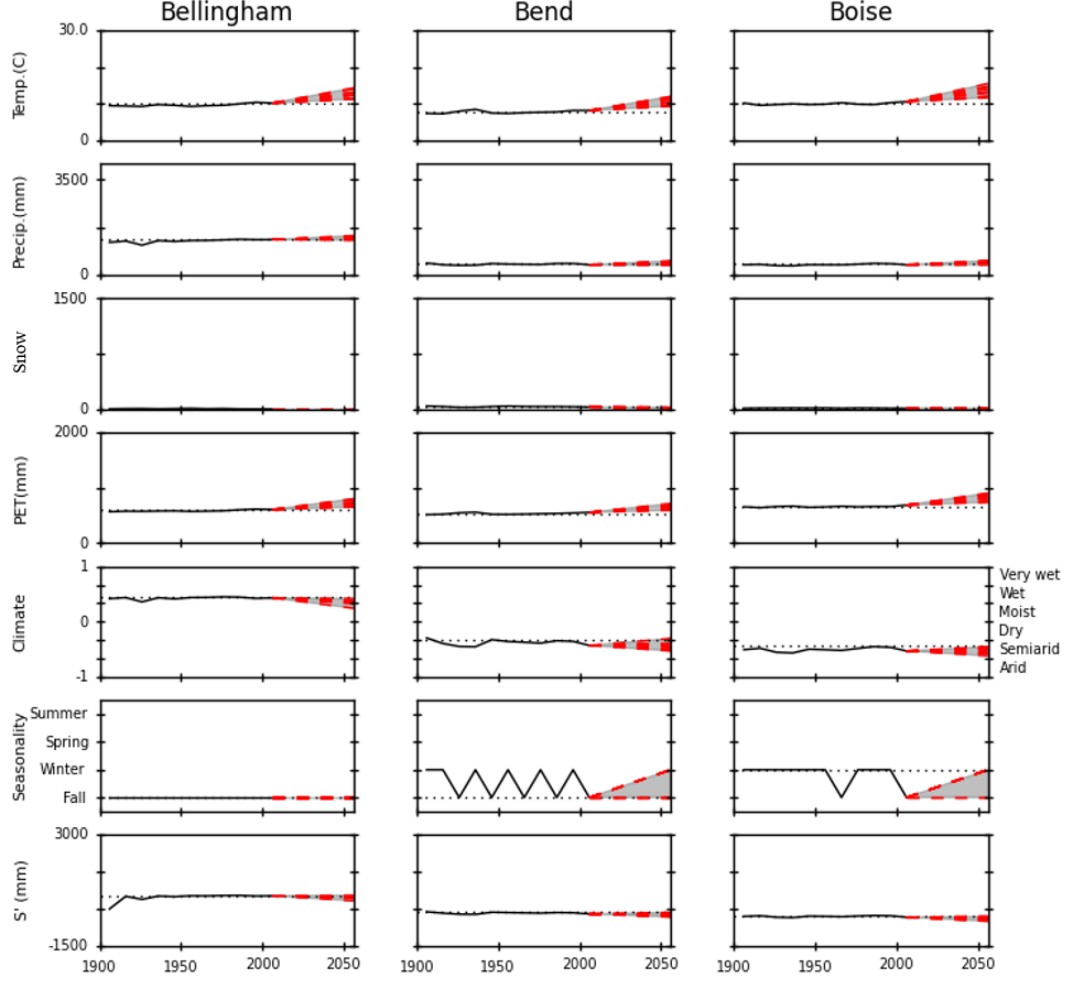






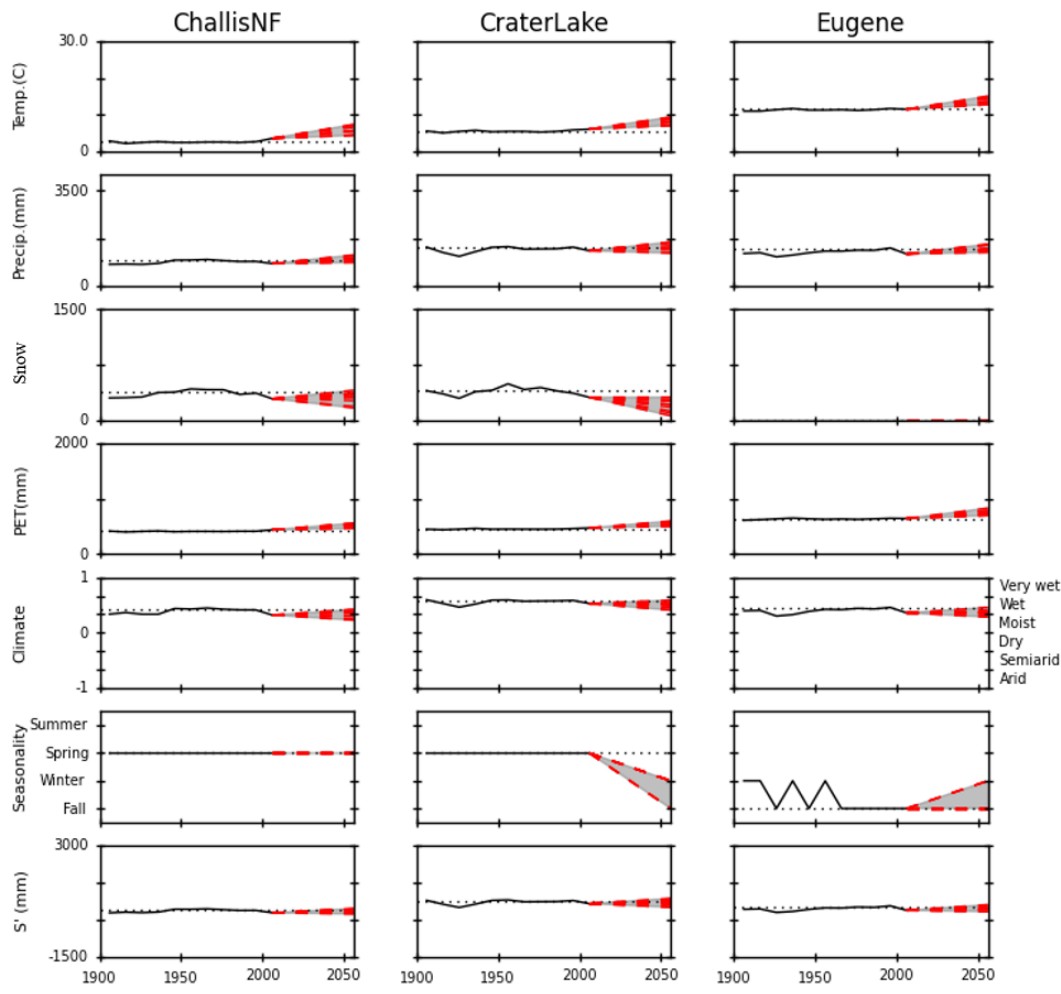




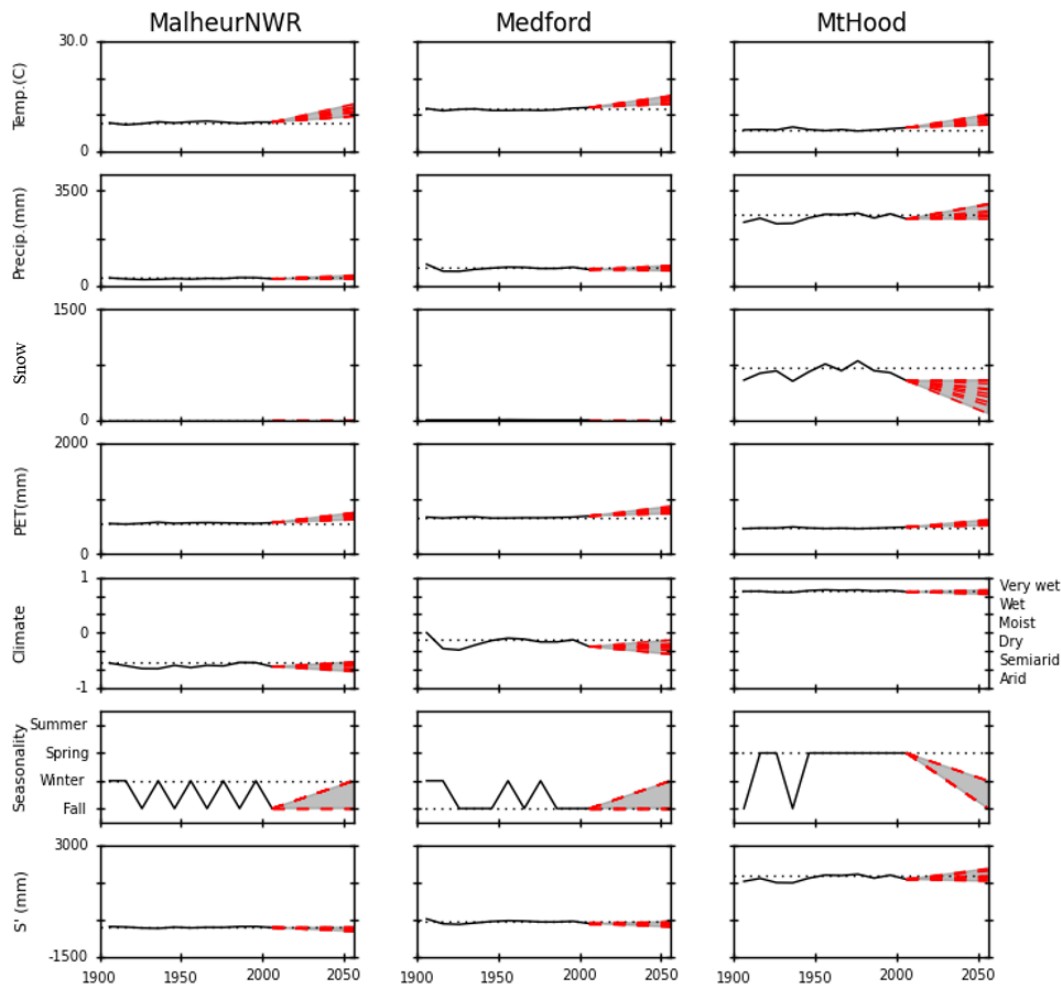






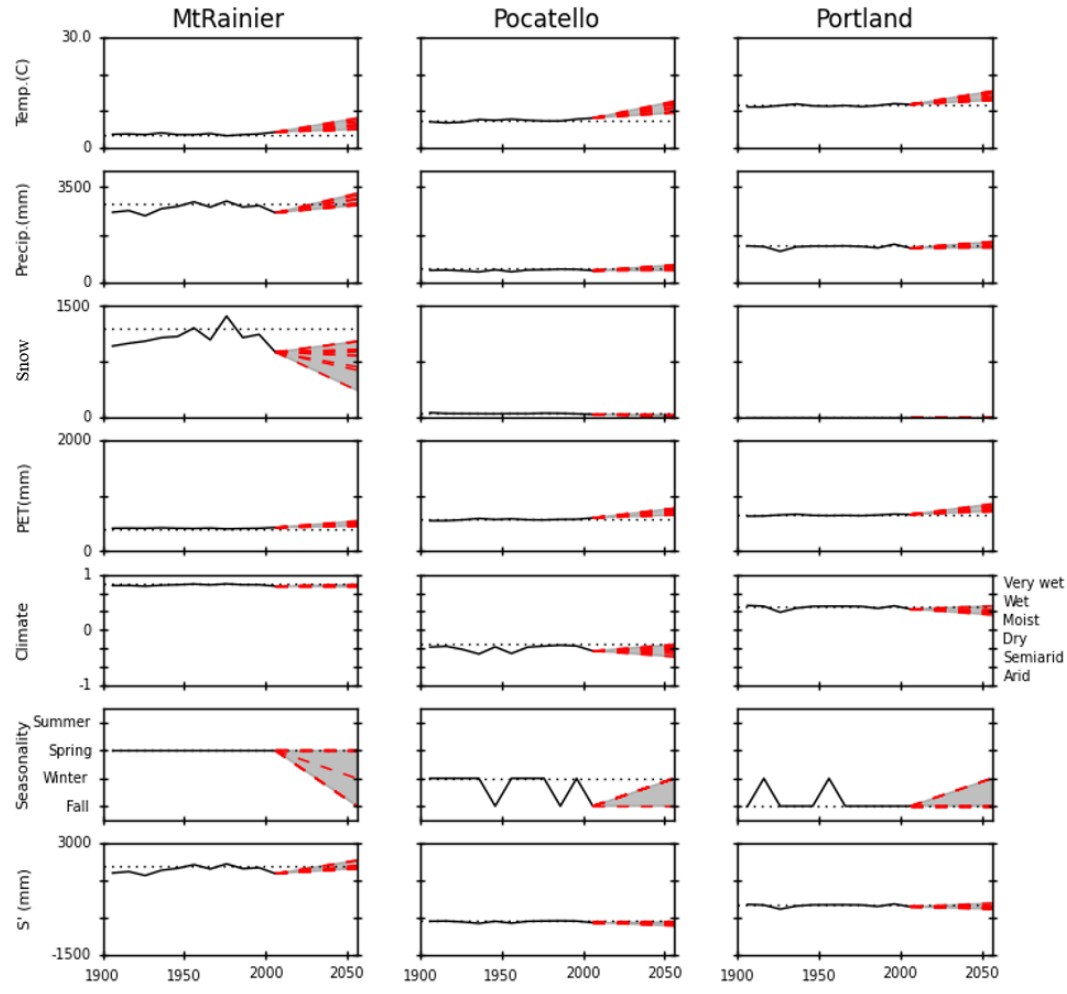




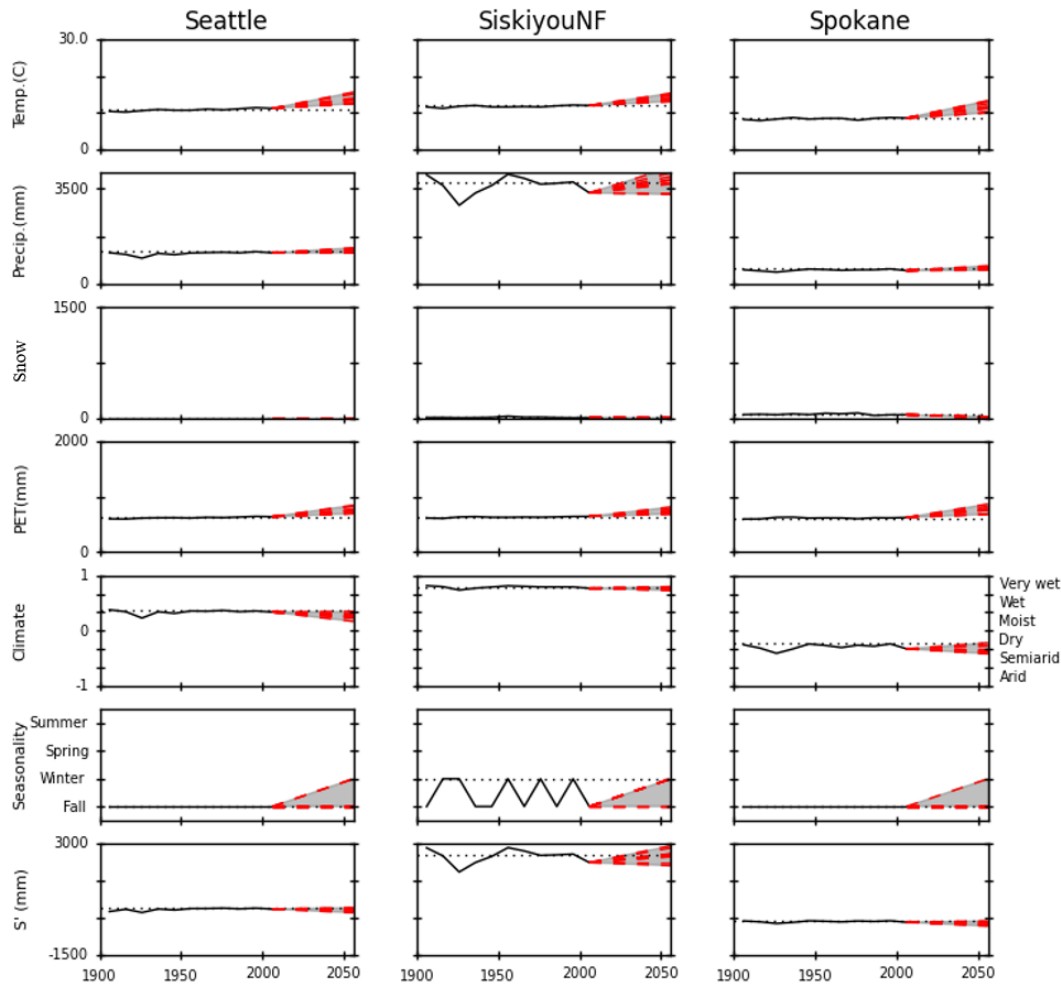






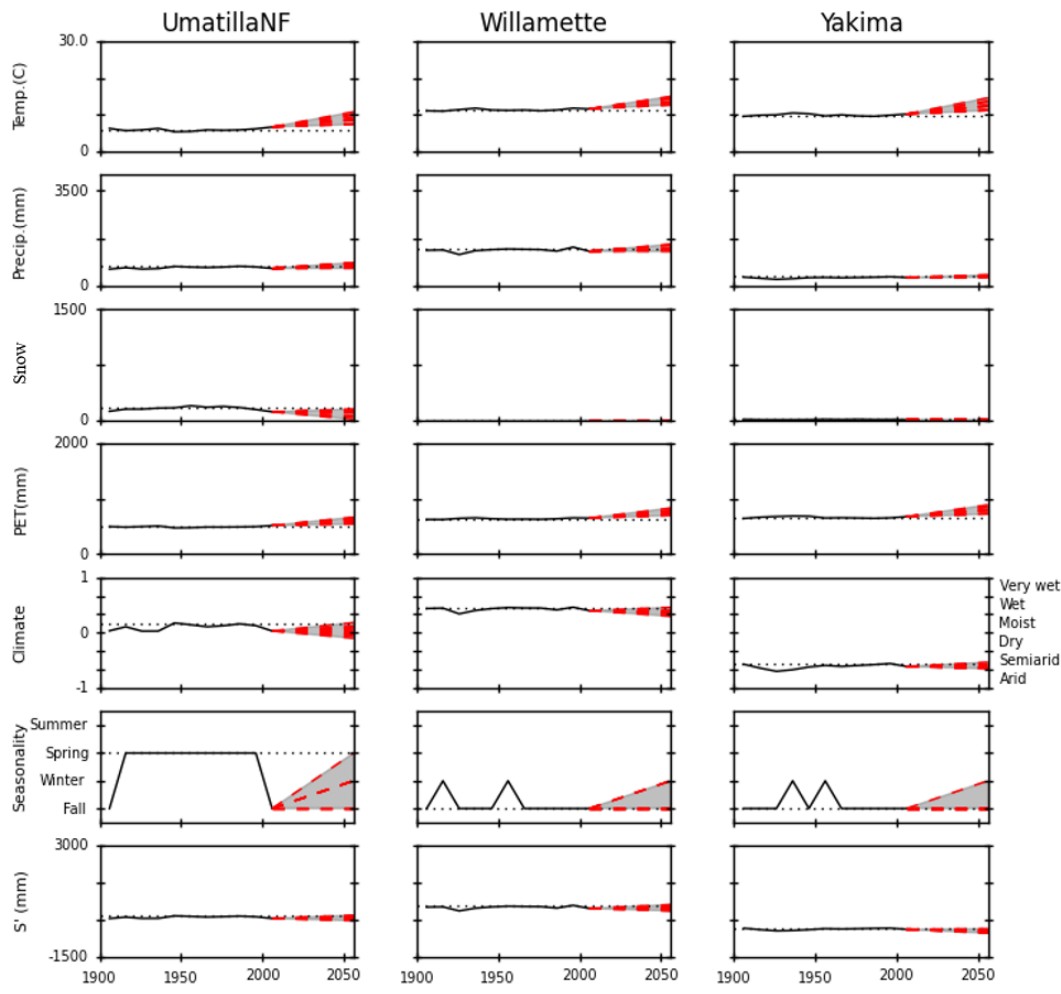




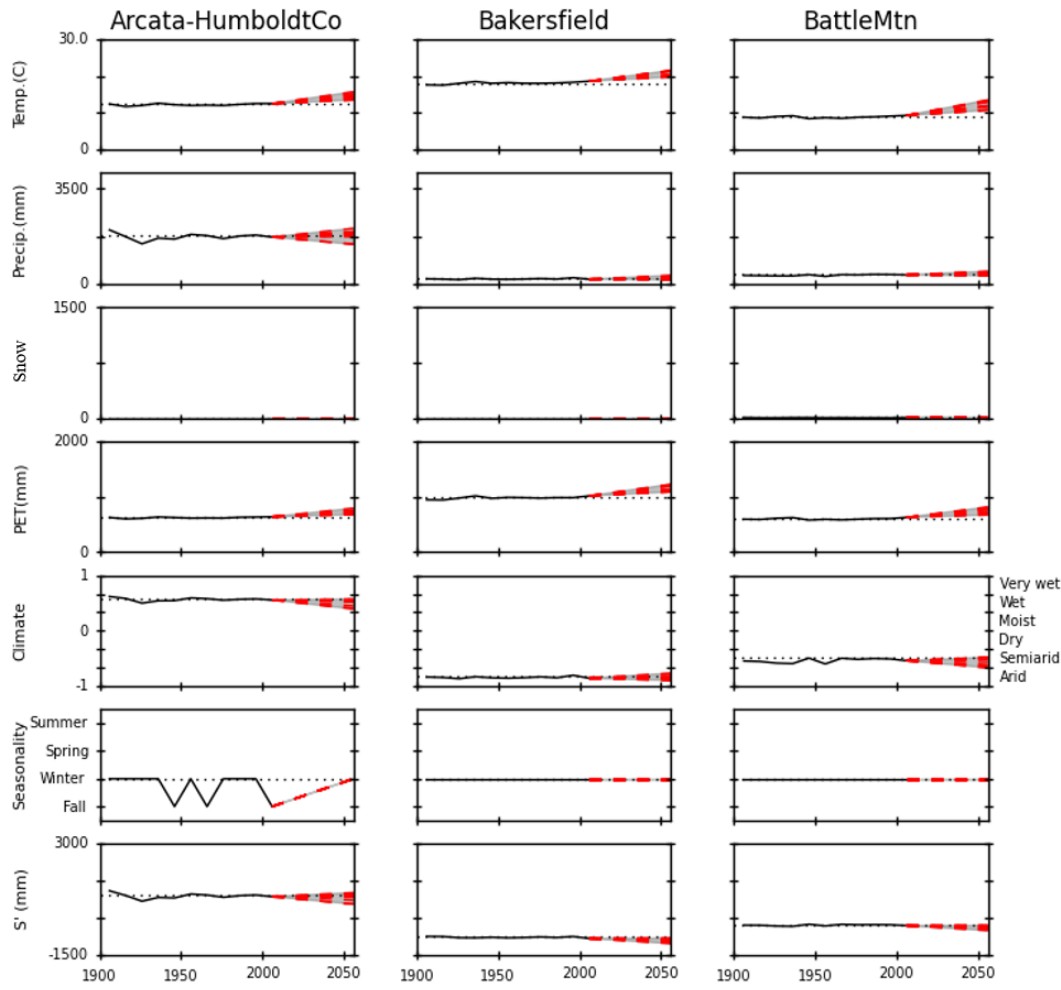




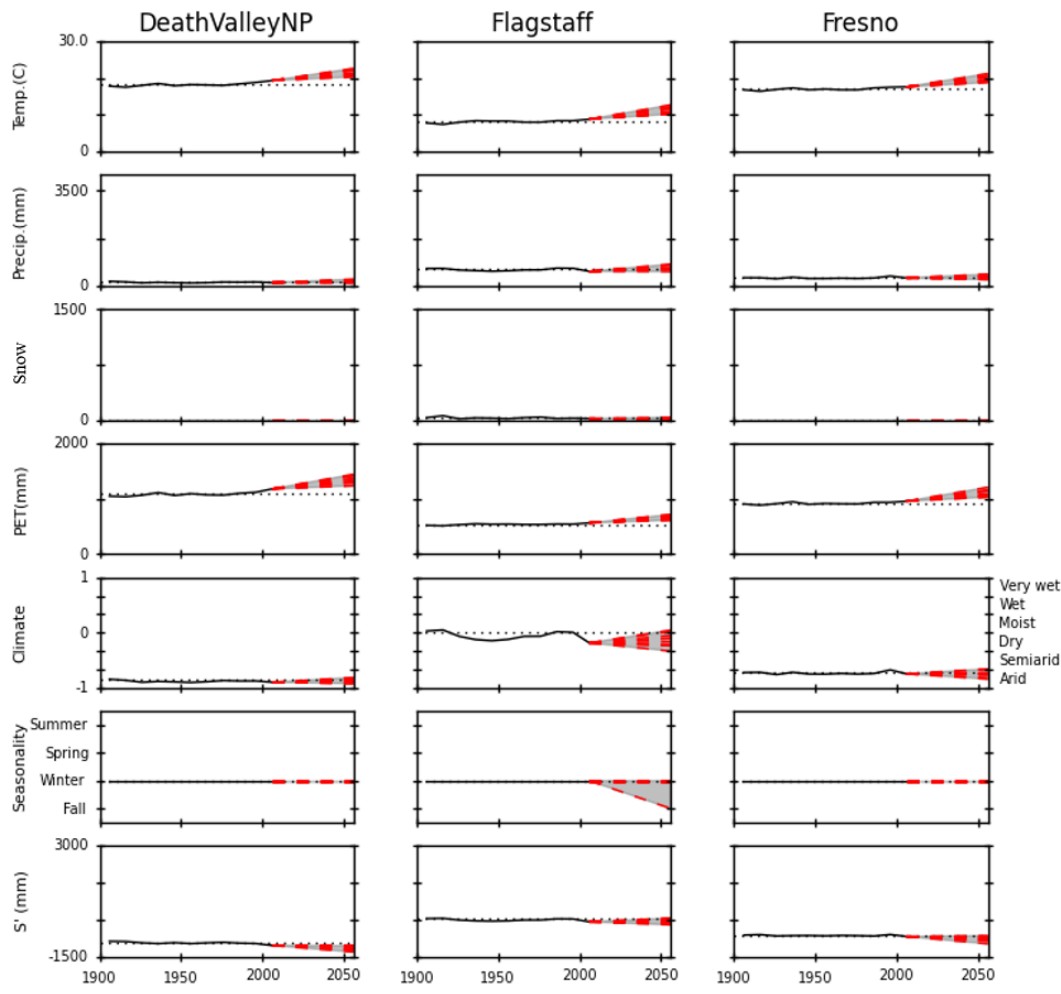




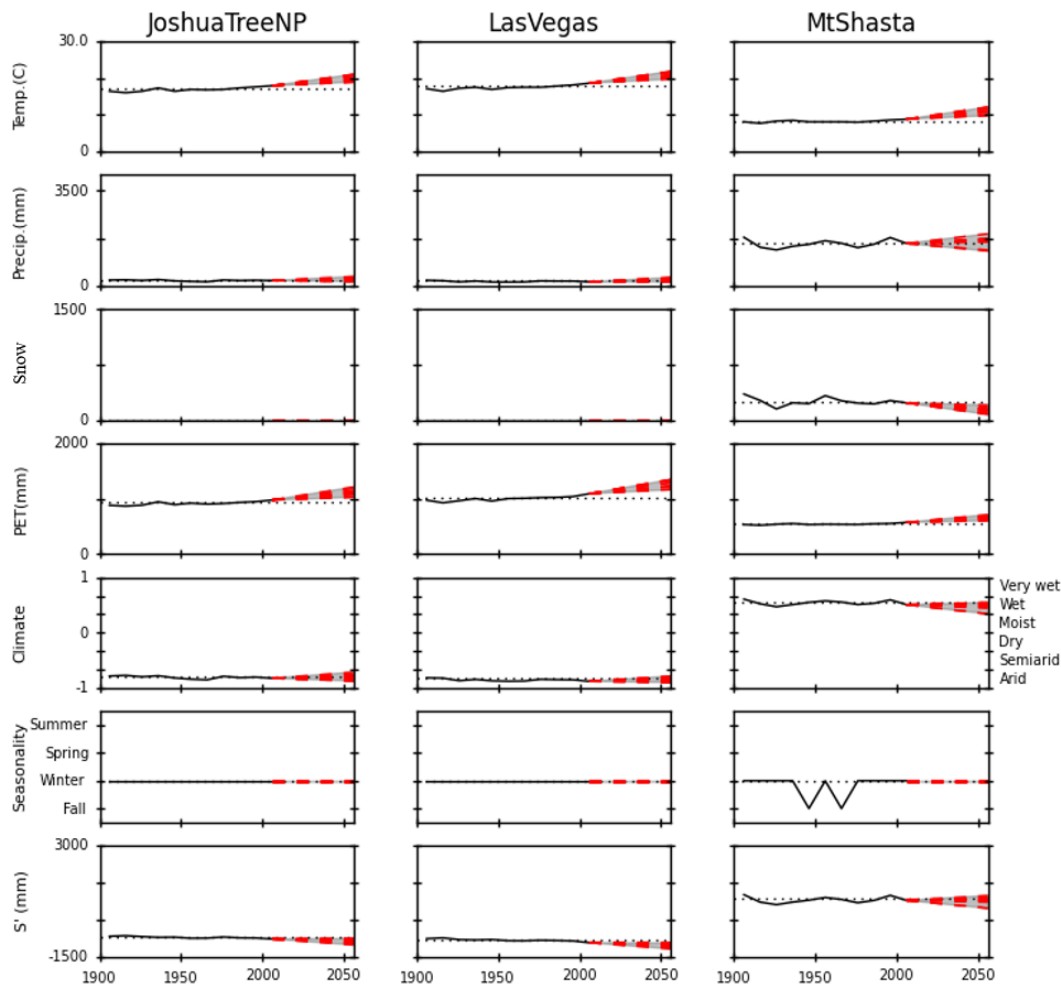






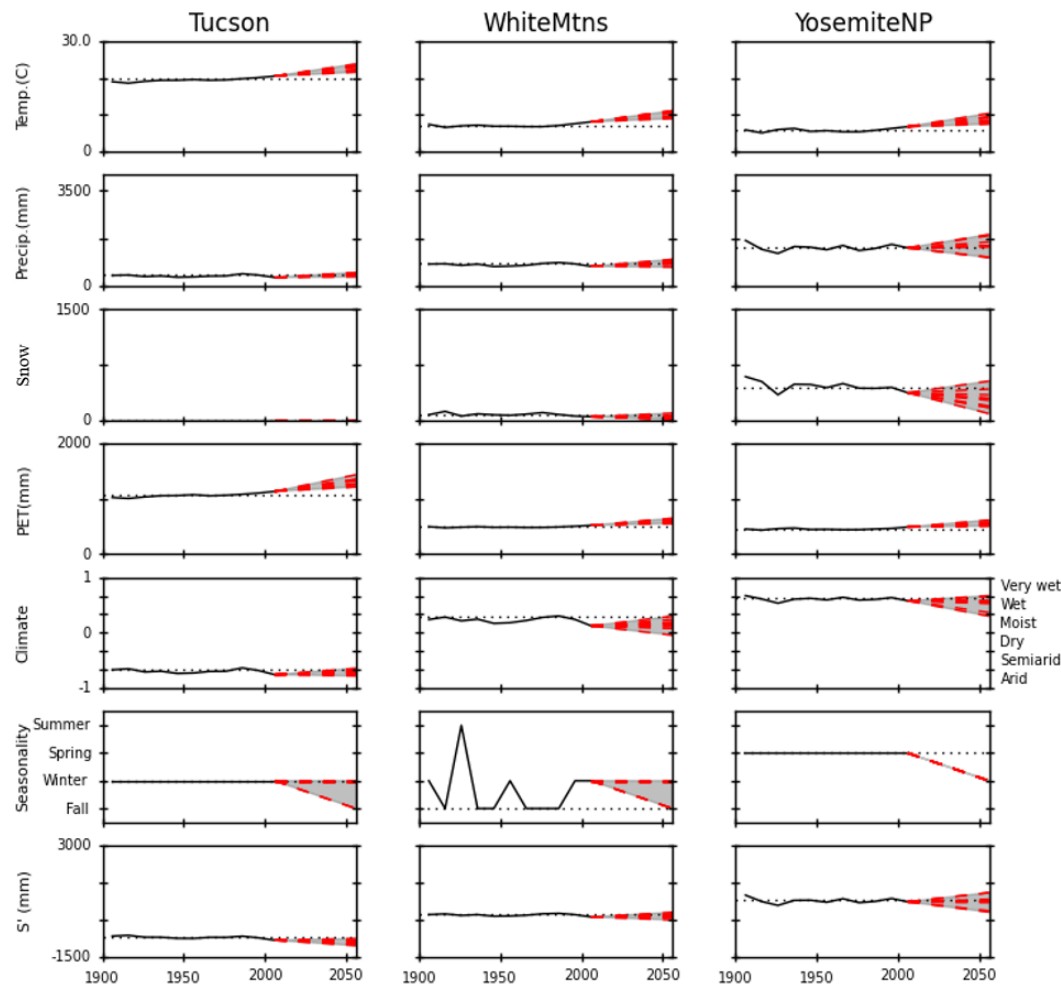




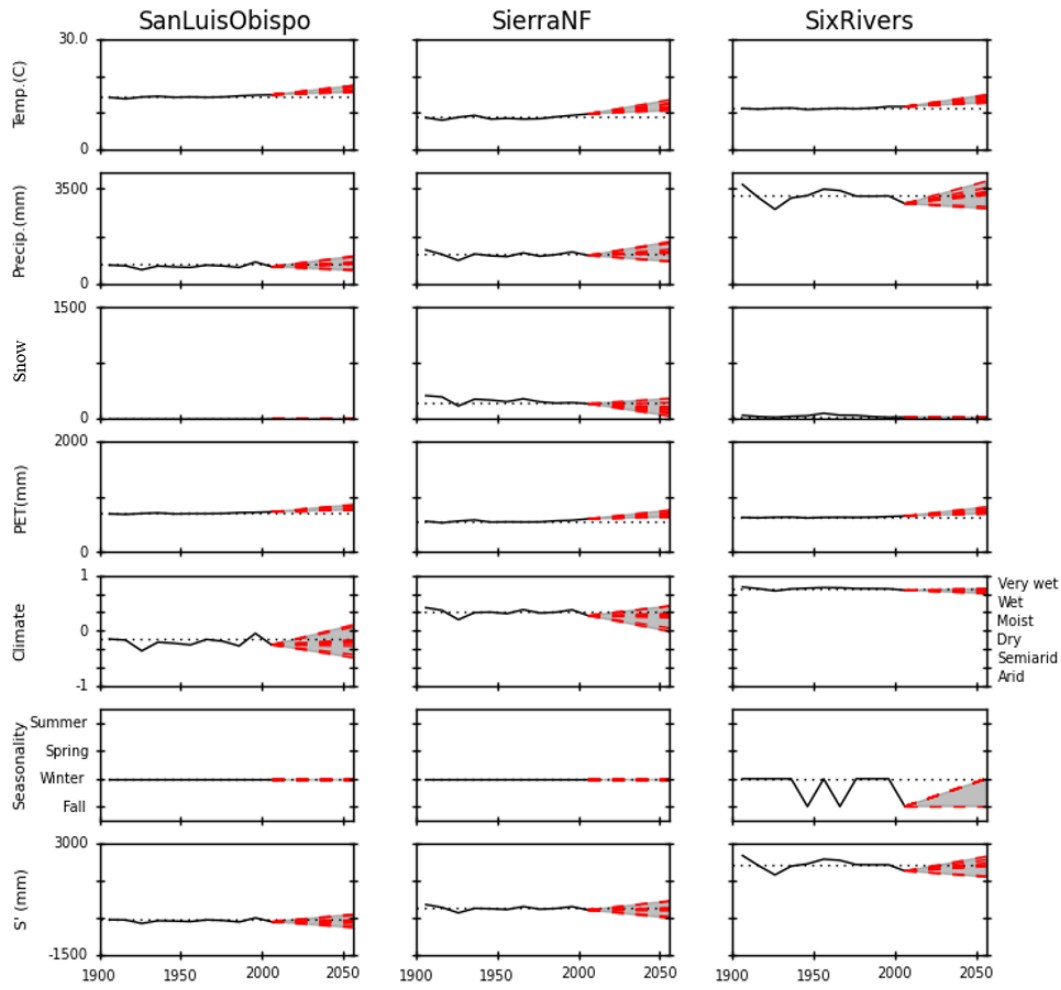






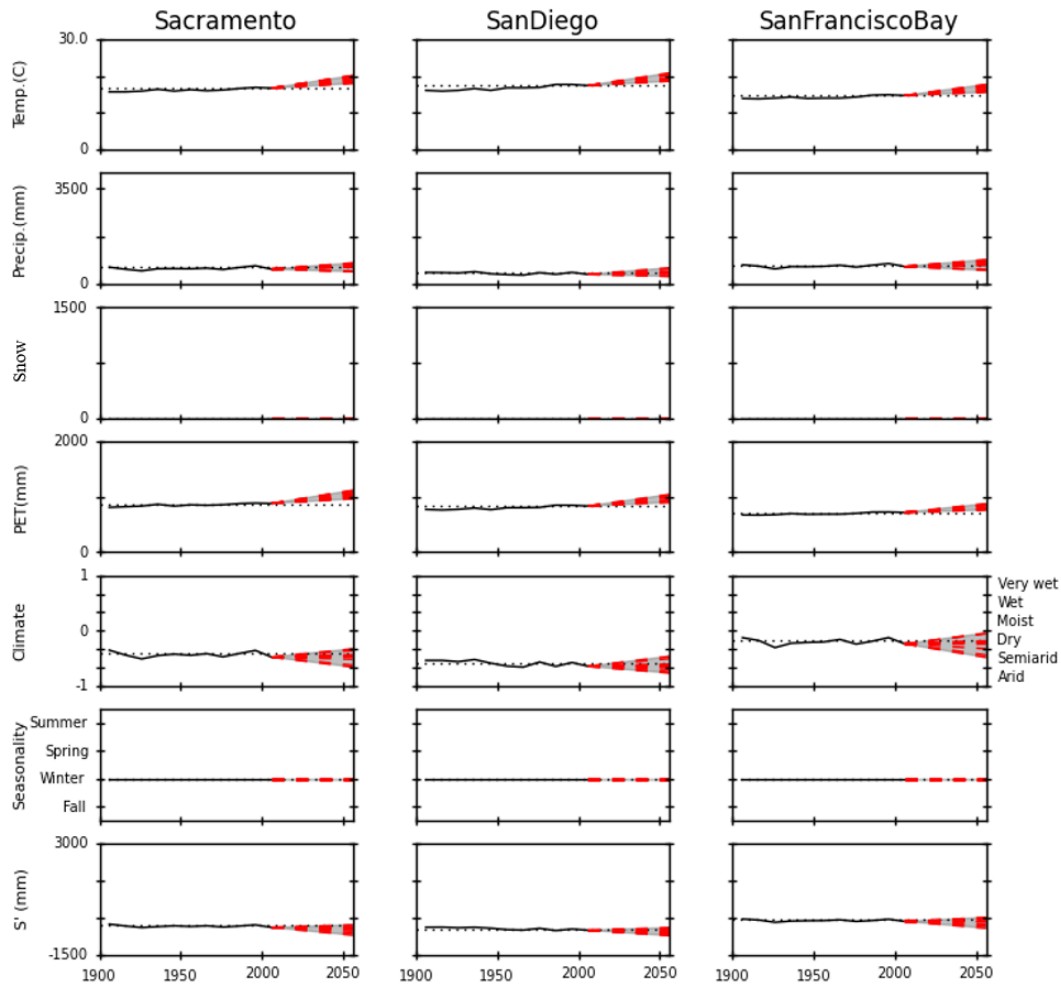






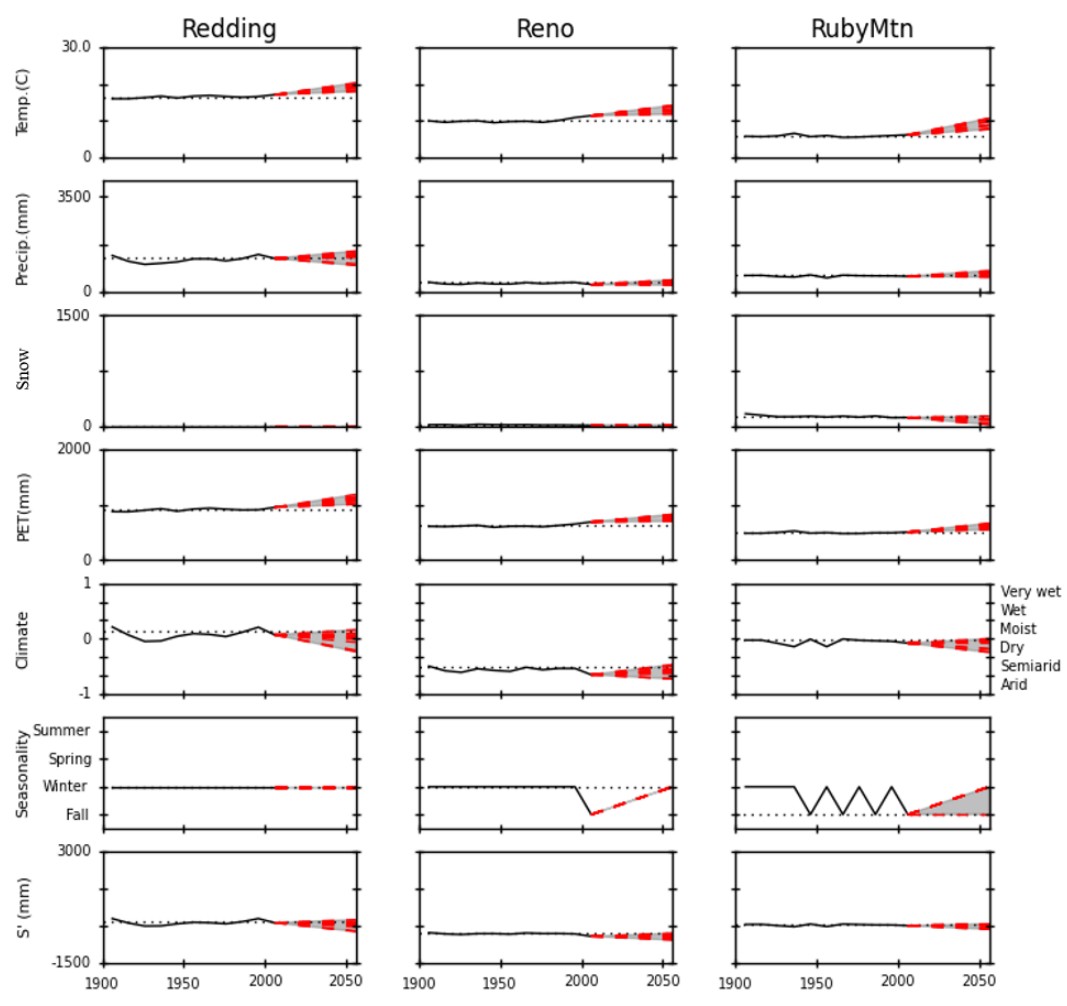

