# Peer review of "Using hydrologic landscape classification and climatic time series"

_Hydrology and Earth System Sciences, 2019_

## Short Comment (SC1) · 22 Mar 2020

I noted that you cited my work describing vulnerability of stream temperatures to climate change. This paper by Sulochan Dhungel and others may be as relevant to this study since they looked specifically at hydrologic changes in response to climate change: https://onlinelibrary.wiley.com/doi/abs/10.1002/rra.3029

---

## Author Comment (AC1) · 23 Mar 2020

Thank you Dr. Hill, We will examine the recommended paper and consider it for inclusion in the revised manuscript.

---

## Referee Comment (RC1) · Shervan Gharari (Referee) · 30 Mar 2020

The manuscript, using existing indices and geospatial datasets, proposes a framework/rule-based decision making on the vulnerability of the Western U.S to future climate change. The manuscript is interesting and encompasses significant data management and GIS work. My general comments:

1- Reading the manuscript, I have a feeling that HESS is not really the right journal for this work. Although interesting work, the manuscript seems to be a report/technical memorandum that is turned into a scientific manuscript. I would suggest this work may be better presented in other engineering or water management journals. This is just

my recommendation on better presenting the work in its context to the right audience. Following that, it is rather difficult to provide a scientific feedback to this work. My feedback remains mostly on the clarification of presentation.

2- The use of English language is very good. The flow of the manuscript is smooth.

3- I am not sure if I really understand the linkage between the hydrological landscape classification and the current manuscript. As the authors mentioned in the introduction, the landscape classification is usually at finer resolution than catchment scale. What the author are doing, is more of clustering or zoning of possible system response to climate change (similar to hydrological modeling approach but with less hydrology as only indices are used). The AU are just a unit where the data is compiled at and this is not really linked to the sub-catchment variability intended landscape classification at catchment level.

4- It seems the authors have a decision tree in mind that they use for classification using the input data. I would suggest the author to provide a schematic of their decision or algorithm that provide readers with better understanding of the method. Similarly, there is no visualization of the shapefile/regions used to create the vulnerability map.

5- I would say the context of vulnerability is missing here. What is it used for? What is the intended motivation behind this vulnerability assessment?

6- The result section is presented very quickly in (few) paragraph(s).

7- The discussion is kind of back to front. It is rather wordy. I would say it can be significantly shorter and focused on the interpretation of the results given the aim of this study.

8- Conclusion session is very vague. I would suggest the authors to come up with few bullet points Conclusions which readers can have as take home message. Also, the discussion, my pervious comment, can evolve along the line of the conclusion (I mean bullet point conclusions can help discussion significantly).

My overall suggestion is to change the manuscript into technical note. I would strongly suggest shortening of the manuscript and remove wordy sections (for example, in discussion). Explain the decision tree visually and elaborate that in methology section. Present the forcing and geospatial data in the decision tree and also visually. I believe major revision is inevitable.

With kind regards,

Shervan Gharari

---

## Referee Comment (RC2) · Anonymous Referee #2 · 6 Apr 2020

This paper aims to apply the hydrologic landscape approach to chronicling changes to the western US under climate change. Using a vulnerability index, the authors aim to highlight locations more or less prone to changes in climate for various indices. It is apparent that the manuscript was a technically challenging effort to reconcile multiple datasets and climate scenarios and synthesize them in a GIS framework. The work there should not be discounted. The manuscript is generally well written, but feels disjointed and an attempt to reconcile several disparate research efforts between a discussion on climate change, hydrologic landscapes, and vulnerability of socially/economically valuable locations. I understand what the authors are aiming for if the authors aim to show all three aspects, they could be better unified. Overall,

I think this paper is worthy of publication and the data analysis is commendable, but better structure and explanation is needed. I recommend the paper be revised and resubmitted.

Specific Comments: L64-65: "The findings are consistent across studies in many areas of the globe...." –How are they consistent across studies and which studies? And the second half of this sentence seems to contradict the first half when you say they aren't consistent without any citations being given. Which one is it?

L175-177: The methods describe that Leibowitz et al (2016) used a modifcation of Wigington et al (2013). It's a little unclear as to what the modification was. More clarity or explanation needed here.

Section 2.3: I'm a little unclear as to the selection of these dates and selection of data. –Why is 1971-2000 considered the modern climate normal when such data is at least 20 years ago? It seems incongruous to have this be your "modern" normal when you consider "historical" data to be up to at least 2010 and state that the PRISM data you use for your calculation of modern normals goes from 1895-present. Why not have the modern normal represent a more recent time period? –I'm also unclear as to why monthly precipitation and mean temperature data is acceptable for the modern climate normal calculation (L230-232), but daily measurements are needed for the historical decadal analyses (L240) and they're subsequently averaged to monthly means anyways. The requirement for daily data caused you to employ a downscaling approach, potentially introducing more error. More explanation is needed here.

Section 2.3.3: Better explanation is needed as to what were the criteria for choosing the 10 modeled emissions scenarios. Figure 2 appropriately shows their distribution in terms of precipitation and temperature, but how were the 10 out of the at least 38 chosen? Random draw? Some other selection criteria? Further, the coloring and subscript numbers in Figure 2 needs to be better explained in the caption.

Section 2.5: Better explanation is needed for how these sites were selected and how

their areal extent was decided (see Table 2). Site specific areal extent appears to range from 38 km2 to 4855 km2 (Table 2). Also according to section 2.2 the target AU size is 80 km2 meaning that at the low end (e.g. Great Basin NP w/ area of 38 km2) is likely composed of a single AU and many only a few units. I get the challenges of making AUs a representative size across multiple different spatial datasets, but some discussion of how that AU size and differential location areal extent affects these location based analyses is warranted.

L312-313: "The time series for the decadal averages for each of the seven HL metrics..." I think you mean to say the seven climate related HL metrics here because things like elevation, subsurface permeability, and surface permeability aren't subject to change under this approach.

L325-327: The sentence beginning "In terms of the 1971-2000 climate normal period" needs some revision. I think it needs a clause saying, "followed by 24% of the area showing fall seasonality, 13% spring seasonality,...."

L342-343: Needs some clarification. What remaining models? You said in methods you only tested 10 and in the preceding text you said 3 may be wetter and 7 generally drier. What models are left?

–Several times in the results and discussion you point out patterns shown along major geographic features like mountain ranges (just as one example paragraph beginning on L351). It would be beneficial to show where those are like you do in the ecoregions in Figure 1. Many of the readers may be unfamiliar with where these features are so it makes it difficult to place the patterns you're describing.

–Several times the authors talk about the sensitivity or vulnerability changes without talking about the direction of that change (but see sentence beginning on L371 "The map for S'" as an example). It'd be useful to make sure if you're saying an area is vulnerable to a change in climate, it's not just the metric (e.g. temperature) but also whether higher/lower or earlier/later.

L385-392: I mentioned in my introduction that the paper was one that seemed disjointed between the paper being primarily one communicating climatic changes with a discussion of HLs and vulnerability thrown into the mix. This is an example of where the authors do a good job of uniting at least the HL approach with the climate indices. The HL approach is the story of the complete code where certain indices may play a more proximal role in given locations. This section does a good job of explaining how the changes in the climate indices could have differential effects based on things like elevation or the permeability metrics. More discussion like this is needed throughout the paper.

Figure 3-4. These colors are hard to discern with much of the area looking a yellow color that according to the sale is no change. I wonder if more of a categorical variable would be appropriate here to show changes rather than a color ramp. The reddish hue is more noticeable in Figure 4 for sure, but I wonder if this could be better communicated.

Figure 7. I wonder if here too classed variables may be better used to show variability by placing then in a low, medium, high type construct rather than a color ramp.

Figure 8. I'm confused by several aspects of this figure. –Several of the figure panels don't have corresponding descriptions in the caption. For instance there's a two separate panels for April 1 SWE and Snow, but the figure caption says "snow (April 1 SWE (mm))" as if they're combined somehow. This is confusing especially when the "Snow" panel has a y-axis labeled 1-2000 without any units. –Also the panel labeled "Climate" I believe is referenced as "FMI" in the caption. Also on this panel the left y-axis is from -1 to 1 while the right axis is the categorical Arid-Very wet labels. This needs to be better explained as it's confusing even to someone familiar with HLs. –Finally, the climate projection section is also confusing as sometimes it appears there's two lines while others there are several (e.g. Mt. Hood SWE panel). It may be cleaner to just show the high and low range lines rather than all the model scenarios I think you're showing. What the gray shaded area is showing also needs to be described in the caption.

[Figure]

Other Discussion Points –The discussion of the site specific locations seems a bit disjointed in the discussion. I wonder if it'd be better served to be called out as case studies in a subsection. The discussion seems to go big picture, dive down to case studies, and then back out to a discussion of HLs. This organization seems a bit haphazard and cobbled together. I wonder if better flow and cohesiveness from section to section could be achieved here.

–The results section is dominated by description of the climatic time series and changes to the HL indices classifications, but explanation of those changes largely disappear in the discussion and is dominated by discussion of vulnerability. I get that the vulnerability index is an attempt to merge some of those ideas, but I would have expected better mixing of the climate and HL information in with the vulnerability discussion.

–You stated in the methods that you chose from the highest emissions scenarios climate data projections (RCP 8.5). Better admonition of that fact needs to be detailed in the discussion as several other projection scenarios show lower degrees of change or better explanation of why you thought the high-end emissions scenarios were most representative needs to be explained.

-I think there could be better discussion as to how having high vulnerability in a single metric could have profound implications in some areas while other areas may only be affected by having high vulnerability across multiple metrics. You get at some of this in the case study approach where certain grape varietals are more impacted by temperature changes say rather than precipitation changes, but I think that could be expressed better throughout including in the discussion of HLs. For instance a change in seasonality could have profound implications to overall hydrology if that change meant a state transition from snow to rain even with a relatively modest change in temperature. There's a robust literature (especially for the west coast) on the impacts of these projected changes. Maybe some incorporation of overall vulnerability across all these indices is warranted. Surely that's industry or stakeholder specific in what they deem

"important" as highlighted in the case studies, but better discussion here may be warranted.

–Along those same lines, you dedicate a lot of space both in terms of figures and text towards changes in seasonality (Figure 4-5) and FMI (Figures 2-3). Some discussion on whether you expect those to be the most consequential HL metrics in this region would be useful.

---

## Author Comment (AC2) · 4 May 2020

Thank you for offering us the opportunity to respond to your comments and feedback on our manuscript titled "Using hydrologic landscape classification and climatic time series to assess hydrologic vulnerability of the Western U.S. to climate". The manuscript was reviewed by one reviewer (RC2) that recommended that the paper be "revised and resubmitted", while you (RC1) suggested changing the manuscript into a technical note. We believe that we can effectively address the specific concerns in an additional revision that will improve the manuscript and make it more worthy of publication in HESS. Overall, we found the reviewer feedback to be insightful and are certain that it

will ultimately benefit the manuscript. Below are our responses to two reviewer comments and one short comment Please note that the specified line numbers reference the original manuscript submitted to HESS.

RC1 Comment 1) The manuscript, using existing indices and geospatial datasets, proposes a framework/rule-based decision making on the vulnerability of the Western U.S to future climate change. The manuscript is interesting and encompasses significant data management and GIS work. My general comments: Reading the manuscript, I have a feeling that HESS is not really the right journal for this work. Although interesting work, the manuscript seems to be a report/technical memorandum that is turned into a scientific manuscript. I would suggest this work may be better presented in other engineering or water management journals. This is just my recommendation on better presenting the work in its context to the right audience. Following that, it is rather difficult to provide a scientific feedback to this work. My feedback remains mostly on the clarification of presentation.

Author Response 1) We are pleased that the reviewer found the manuscript interesting, although we respectfully disagree with their suggestion that this research would be more appropriate for an engineering or water management journal. There were few concrete reasons provided by RC1 to justify such a conclusion, especially given that, by their own admission, they provided little scientific feedback other than on presentation. This was in direct contrast to RC2, who provided specific critical feedback that was helpful in improving the manuscript.

More specifically, according to HESS's website: "HESS encourages and supports fundamental and applied research that advances the understanding of hydrological systems, their role in providing water for ecosystems and society, and the role of the water cycle in the functioning of the Earth system." We feel that this manuscript is very well suited to HESS as our research uses an integrated approach to examine how hydrologic function drives the vulnerability of certain geographies and socio-economic industries to climatic impacts. We analyzed how basic hydrologic function might change

across the Western U.S. under future climate predictions. We applied a hydrologic vulnerability index across the entire landscape and then examined how the vulnerability indices could be applicable to economically important industries but could also be informative about how the impacts of climate change may impact ecosystems and society at landscape scales.

Thus, we believe that HESS has the ideal audience for this research. This work is much less suited for a water management or engineering journal, as we do not provide research that is very relevant to water management studies or water resource engineering. We are also not presenting research about decision management tools or water management.

RC1 Comment 2) The use of English language is very good. The flow of the manuscript is smooth.

Author Response 2) Thank you.

RC1 Comment 3) I am not sure if I really understand the linkage between the hydrological landscape classification and the current manuscript. As the authors mentioned in the introduction, the landscape classification is usually at finer resolution than catchment scale. What the author are doing, is more of clustering or zoning of possible system response to climate change (similar to hydrological modeling approach but with less hydrology as only indices are used). The AU are just a unit where the data is compiled at and this is not really linked to the sub-catchment variability intended landscape classification at catchment level.

Author Response 3) We respectfully disagree with the reviewer's interpretation of the Hydrologic Landscape approach with respect to two major issues. First, Hydrologic Landscapes have never been applied at scales finer than at what we define as the catchment scale. In fact, Tom Winter, who originally developed the Hydrologic Landscapes concept in 2001, distinguished between six types of generalized Hydrologic Landscapes, which were conceptualized as large landscape units. When these were

originally mapped by Wolock et al. in 2004, it resulted in 20 non-contiguous regions over the entire US, which were much larger than the catchment scale. The Hydrologic Landscape approach applied in the current application are at a much finer resolution than the Wolock et al. maps, yet are still a comparative analysis between catchments, and not an analysis within catchments. Secondly, while the original Wolock et al. Hydrologic Landscape analysis was in fact a clustering approach, the current analysis, which derived from the approach championed by Wigington et al. 2013, is not. In fact, the Wigington et al. approach specifically chose not to use a clustering approach, and instead used a conceptual approach that selected a priori important factors known to contribute to hydrologic flow. Furthermore, more recent studies using the hydrologic landscape classification approach have been applied at a subcatchment scale (Leibowitz et al. 2014, Patil et al. 2014, Leibowitz et al. 2016, Todd et al. 2017), using a similar approach as used in this analysis. In fact, we use the same subcatchments used in the Leibowitz et al. 2013 study and applied their method to delineate the subcatchments for our study's expanded 6 state study region. We could add further clarification of this process, however, in favor of brevity, we prefer citing the methods used in previous studies, which we simply applied to our expanded study area.

RC1 Comment 4) It seems the authors have a decision tree in mind that they use for classification using the input data. I would suggest the author to provide a schematic of their decision or algorithm that provide readers with better understanding of the method. Similarly, there is no visualization of the shapefile/regions used to create the vulnerability map.

Author Response 4) We can explore the idea of developing a graphical depiction of the hydrologic landscape classification approach, although the classification specifics are described in the methods: FMI (L187-188); Seasonality (L201-203); Subsurface Permeability (L211); Terrain (L217-219); and Surface Permeability (L225-227). We can also further clarify that the vulnerability maps depict the ∼24,000 AUs that were classified for each Vulnerability parameter, since it wouldn't be very helpful to create a
map that shows the ∼24,000 AUs.

RC1 Comment 5) I would say the context of vulnerability is missing here. What is it used for? What is the intended motivation behind this vulnerability assessment?

Author Response 5) The vulnerability index can be used to identify areas that are most likely to experience altered hydrologic conditions in the future relative to the 1900-2010 period. The intended motivation behind this vulnerability assessment is to help stakeholders determine how likely they are to experience changes to one or more hydrologic parameters that are important for them or a particular industry / application.

RC1 Comment 6) (Section 3: Results): The result section is presented very quickly in (few) paragraph(s).

Author Response 6) The text portion of our results section occupies 8 paragraphs, which equates to 3 out of 14 pages or 1800 out of 8091 words (22% of the text of the primary manuscript). This does not include tables or figures that will ultimately be placed within the section. We believe this provides a section that is thorough and sufficiently summarizes the results. This seems relatively reasonable and we wouldn't want to expand it further without feedback on specific deficiencies or direction from the editor.

RC1 Comment 7) Section 4: Discussion): The discussion is kind of back to front. It is rather wordy. I would say it can be significantly shorter and focused on the interpretation of the results given the aim of this study.

Author Response 7) This comment would be challenging to address as it lacks specificity. We can certainly examine the content of the discussion and look for opportunities to be more brief, while maintaining clarity. We will also need to balance this comment with relevant comments by the other reviewer. That reviewer specifically suggested adding a subsection to the discussion to add clarity, which was helpful and should ultimately improve the paper.

[Figure]

RC1 Comment 8) (Section 5: Conclusions): Conclusion session is very vague. I would suggest the authors to come up with few bullet points Conclusions which readers can have as take home message. Also, the discussion, my pervious comment, can evolve along the line of the conclusion (I mean bullet point conclusions can help discussion significantly).

Author Response 8) It seems likely that the conclusions section will change as the manuscript evolves. We can highlight our take home messages. We can also examine whether there is an opportunity to utilize bullet point summaries in the discussion.

RC1 Comment 9) My overall suggestion is to change the manuscript into technical note. I would strongly suggest shortening of the manuscript and remove wordy sections (for example, in discussion). Explain the decision tree visually and elaborate that in methology section. Present the forcing and geospatial data in the decision tree and also visually. I believe major revision is inevitable.

Author Response 9) We can attempt to shorten the discussion section, which at present is 1615 out of 8091 words (20% of the manuscript), although we note that this is slightly shorter than our results section which the Reviewer thought was much too short. We would also prefer to shorten the paper overall (which aligns with this reviewer's feedback) and will look for opportunities to do so. We can also look for opportunities to summarize the methods visually with a flow chart (as a decision tree wouldn't quite be a proper fit to our analyses).

---

## Author Comment (AC3) · 4 May 2020

Thank you for offering us the opportunity to respond to your (RC2) comments and feedback on our manuscript titled "Using hydrologic landscape classification and climatic time series to assess hydrologic vulnerability of the Western U.S. to climate". We believe that we can effectively address the specific concerns in an additional revision that will improve the manuscript and make it more worthy of publication in HESS. We found the reviewer feedback to be insightful and are certain that it will ultimately benefit the manuscript.

RC2 Comment 1) This paper aims to apply the hydrologic landscape approach to

chronicling changes to the western US under climate change. Using a vulnerability index, the authors aim to highlight locations more or less prone to changes in climate for various indices. It is apparent that the manuscript was a technically challenging effort to reconcile multiple datasets and climate scenarios and synthesize them in a GIS framework. The work there should not be discounted. The manuscript is generally well written, but feels disjointed and an attempt to reconcile several disparate research efforts between a discussion on climate change, hydrologic landscapes, and vulnerability of socially/economically valuable locations. I understand what the authors are aiming for. If the authors aim to show all three aspects, they could be better unified. Overall, I think this paper is worthy of publication and the data analysis is commendable, but better structure and explanation is needed. I recommend the paper be revised and resubmitted.

Author Response 1) Thank you for this feedback and recognition of the value of the research effort. RC2 recognizes that a strength of the manuscript is our attempt to integrate three disparate fields of 1) climate change, 2) Hydrologic Landscape classification, and 3) the socioeconomic impacts of climate change. We propose that we can highlight this strength of the analysis in the introduction which will also help us unify these aspects of the study. We propose to add some language of this unifying concept into the Introduction and then revisit the unifying concept in the closing paragraphs of the Discussion or Conclusion sections.

Specific Comment 2) (Lines 64-65): "The findings are consistent across studies in many areas of the globe...." –How are they consistent across studies and which studies? And the second half of this sentence seems to contradict the first half when you say they aren't consistent without any citations being given. Which one is it?

Author Response 2) This sentence is referring to McAfee's 2013 study and was intended to summarize her findings. We will modify the text to clarify the study's results.

Specific Comment 3 (Line 175-177): The methods describe that Leibowitz et al (2016)

used a modification of Wigington et al (2013). It's a little unclear as to what the modification was. More clarity or explanation needed here.

Author Response 3) We can add text to further summarize Leibowitz et al. (2016)'s modification to the Wigington et al. (2013) methods.

Specific Comment 4 (Section 2.3): I'm a little unclear as to the selection of these dates and selection of data. Why is 1971-2000 considered the modern climate normal when such data is at least 20 years ago? It seems incongruous to have this be your ""modern"" normal when you consider "historical" data to be up to at least 2010 and state that the PRISM data you use for your calculation of modern normals goes from 1895-present. Why not have the modern normal represent a more recent time period?

Author Response 4) We agree that the use of the term "modern" is inaccurate and we will remove it. We chose to use the 1971-2000 period because the analysis was intended to complement the Leibowitz et al. 2016 study, which used 1971-2000 as its defined "climate normal". We can add explanatory text to our reasoning for defining our normal climate period as 1971-2000.

Specific Comment 5 (Section 2.3; Lines 230-240): I'm also unclear as to why monthly precipitation and mean temperature data is acceptable for the modern climate normal calculation (L230-232), but daily measurements are needed for the historical decadal analyses (L240) and they're subsequently averaged to monthly means anyways. The requirement for daily data caused you to employ a downscaling approach, potentially introducing more error. More explanation is needed here.

Author Response 5) While we alluded to this detail on line 241 in the original manuscript, we had previously acquired the 1971-2000 400m monthly climate normals for a fee. Budget constraints did not allow us to purchase the remaining decadal data at 400m monthly resolutions. However, the 4km resolution daily historical data was available at no cost. Therefore, we chose to utilize the freely available data in our analyses. The daily 4km data were used to generate mean monthly gridded datasets and

were downscaled to match the scale and resolution of the 400m mean monthly precipitation and temperature datasets. While we expected that the daily data would have greater accuracy overall, we felt that both datasets should be comparable at monthly time-scales. We can add explanatory text that clarifies the reasons for these decisions.

Specific Comment 6 (Lines 262-265 and Fig. 2): Better explanation is needed as to what were the criteria for choosing the 10 modeled emissions scenarios. Figure 2 appropriately shows their distribution in terms of precipitation and temperature, but how were the 10 out of the at least 38 chosen? Random draw? Some other selection criteria? Further, the coloring and subscript numbers in Figure 2 needs to be better explained in the caption.

Author Response 6) We can add clarifying language to lines 262-265 regarding the model simulation selection process. We can also add clarifying information about the figure coloring and naming conventions to the caption of Fig. 2.

Specific Comment 7 (Section 2.5; Lines 309-316): Better explanation is needed for how these sites were selected and how their areal extent was decided (see Table 2). Site specific areal extent appears to range from 38 km2 to 4855 km2 (Table 2). Also according to section 2.2 the target AU size is 80 km2 meaning that at the low end (e.g. Great Basin NP w/ area of 38 km2) is likely composed of a single AU and many only a few units. I get the challenges of making AUs a representative size across multiple different spatial datasets, but some discussion of how that AU size and differential location areal extent affects these location based analyses is warranted.

Author Response 7) Specific focus sites were selected subjectively so that we could examine climate impacts at locations that may be of general interest. In addition, the range of Assessment Unit (AU) areas represents watersheds that are larger than hillslopes but smaller than large basins. We can add explanatory language to explain that background information. We can also explain that all of the AUs that had greater than 50% of their area within the geographic boundary of a location were included in the AU
analysis for each location. For instance, the Great Basin National Park (GBNP) was covered by a single AU, rather than numerous AUs because less than 49% of the area contained by the other AU areas fell within the GBNP boundary.

Specific Comment 8 (Lines 312-313): "The time series for the decadal averages for each of the seven HL metrics..." I think you mean to say the seven climate related HL metrics here because things like elevation, subsurface permeability, and surface permeability aren't subject to change under this approach.

Author Response 8) Good catch. We will add the 'climate-related' descriptor to our reference to the seven HL metrics.

Specific Comment 9 (Lines 325-327): The sentence beginning "In terms of the 1971-2000 climate normal period" needs some revision. I think it needs a clause saying, "followed by 24% of the area showing fall seasonality, 13% spring seasonality,. "

Author Response 9) Absolutely. That is an awkward sentence that needs to be revised. Thank you for noting that the sentence structure needs to be improved.

Specific Comment 10 (Lines 342-343): Needs some clarification. What remaining models? You said in methods you only tested 10 and in the preceding text you said 3 may be wetter and 7 generally drier. What models are left?

Author Response 10) Thank you for noticing that duplication of information that basically repeated the information with different wording. We will delete the duplicative text (L340-343).

Specific Comment 11 (Line 355): Several times in the results and discussion you point out patterns shown along major geographic features like mountain ranges (just as one example paragraph beginning on L351). It would be beneficial to show where those are like you do in the ecoregions in Figure 1. Many of the readers may be unfamiliar with where these features are so it makes it difficult to place the patterns you're describing.

Author Response 11) In this paragraph, we do reference the White Mountains (L355),

which is Location #42 in Fig. 1 and table 2. In any case that we have referenced place names that correlate to Locations identified in Table 2 or Fig. 1, we can reference the assigned Location # as well. We also refer to the "Sierra-Nevada Mountains", "Cascade Mountains", and "Mountainous areas in Idaho". We could also address this familiarity issue by either deleting these references in the manuscript; adding a supplemental figure; or adding additional information to Figure 1.

Specific Comment 12 (Line 371): Several times the authors talk about the sensitivity or vulnerability changes without talking about the direction of that change (but see sentence beginning on L371 "The map for S'" as an example). It'd be useful to make sure if you're saying an area is vulnerable to a change in climate, it's not just the metric (e.g. temperature) but also whether higher/lower or earlier/later.

Author Response 12) While it is possible to talk about direction of change (higher or lower than the two standard deviations) for the projection of an individual climate model, the vulnerability index is the integration of ten individual models. It is possible for individual models to exceed the two standard deviation threshold from the mean in both the upper and lower directions; thus there is not a unique direction of change associated with our vulnerability index as we've defined it. We can add text to the methods and results that clarifies this detail of our Vulnerability Index.

Specific Comment 13 (Line 385-392): I mentioned in my introduction that the paper was one that seemed disjointed between the paper being primarily one communicating climatic changes with a discussion of HLs and vulnerability thrown into the mix. This is an example of where the authors do a good job of uniting at least the HL approach with the climate indices. The HL approach is the story of the complete code where certain indices may play a more proximal role in given locations. This section does a good job of explaining how the changes in the climate indices could have differential effects based on things like elevation or the permeability metrics. More discussion like this is needed throughout the paper.

Author Response 13) Thank you for pointing out that the integration of the HL approach with the climate indices is a unique aspect of our manuscript and is worth expanding upon. We can add more text or otherwise add additional HL context to our discussion of climate and the associated socio-economic implications to the introduction, discussion, and/or conclusions.

Specific Comment 14 (Figures 3 & 4): These colors are hard to discern with much of the area looking a yellow color that according to the sale is no change. I wonder if more of a categorical variable would be appropriate here to show changes rather than a color ramp. The reddish hue is more noticeable in Figure 4 for sure, but I wonder if this could be better communicated.

Author Response 14) Figures 3 and 4 do illustrate the actual geographic differences in FMI across large regions. When mapping the differences categorically, the differences either appear exaggerated or absent. Thus, we would prefer to retain these figures in their current form, as we consider categorical differences to be less inaccurate and possibly misleading.

Specific Comment 15 (Figure 7.1): I wonder if here too classed variables may be better used to show variability by placing then in a low, medium, high type construct rather than a color ramp.

Author Response 15) These images do depict classified variable (10 classes), however, the legend suggests that this is a continuous variable. We will modify the legend to clarify that this depicts a classified variable. We would prefer to retain the figures in their current form, as we consider that depicting vulnerability into fewer classes would be less accurate and possibly misleading.

Specific Comment 16 (Figure 8): I'm confused by several aspects of this figure. – Several of the figure panels don't have corresponding descriptions in the caption. For instance there's a two separate panels for April 1 SWE and Snow, but the figure caption says "snow (April 1 SWE (mm))" as if they're combined somehow. This is confusing

especially when the "Snow" panel has a y-axis labeled 1-2000 without any units. Also the panel labeled "Climate" I believe is referenced as "FMI" in the caption. Also on this panel the left y-axis is from -1 to 1 while the right axis is the categorical Arid-Very wet labels. This needs to be better explained as it's confusing even to someone familiar with HLs. –Finally, the climate projection section is also confusing as sometimes it appears there's two lines while others there are several (e.g. Mt. Hood SWE panel). It may be cleaner to just show the high and low range lines rather than all the model scenarios I think you're showing. What the gray shaded area is showing also needs to be described in the caption.

Author Response 16) Thank you for the attention to detail. There was an error in the labeling of one of the figures. The "Snow" figures should be labeled as "PET". We can add clarification to the Climate / FMI panel of figures. We will also explore removing the red dashed lines that illustrate the individual climate model outputs, and only including the range of projections.

Specif ic Comment 17 (Lines 426-521): The discussion of the site specific locations seems a bit disjointed in the discussion. I wonder if it'd be better served to be called out as case studies in a subsection. The discussion seems to go big picture, dive down to case studies, and then back out to a discussion of HLs. This organization seems a bit haphazard and cobbled together. I wonder if better flow and cohesiveness from section to section could be achieved here.

Author Response 17) We can add a subsection to the discussion for case studies and make the discussion section more cohesive.

Specific Comment 18 (Sections 4 and 5: Results and Discussion): The results section is dominated by description of the climatic time series and changes to the HL indices classifications, but explanation of those changes largely disappear in the discussion and is dominated by discussion of vulnerability. I get that the vulnerability index is an attempt to merge some of those ideas, but I would have expected better mixing of the

climate and HL information in with the vulnerability discussion.

Author Response 18) We believe that some of the above proposed changes will address this issue in a revised version. We propose to further highlight the integration of climate, the HL classification approach, climate vulnerability, and socio-economic impacts throughout the paper, especially the intro, discussion, and conclusions.

Specific Comment 19 (Lines 260-262): You stated in the methods that you chose from the highest emissions scenarios climate data projections (RCP 8.5). Better admonition of that fact needs to be detailed in the discussion as several other projection scenarios show lower degrees of change or better explanation of why you thought the high-end emissions scenarios were most representative needs to be explained.

Author Response 19) Good idea. We can add emphasis to the discussion so that it is clear why we chose to analyze and present only results and implications that relate to the RCP 8.5 pathway.

Specific Comment 20 (Other): I think there could be better discussion as to how having high vulnerability in a single metric could have profound implications in some areas while other areas may only be affected by having high vulnerability across multiple metrics. You get at some of this in the case study approach where certain grape varietals are more impacted by temperature changes say rather than precipitation changes, but I think that could be expressed better throughout including in the discussion of HLs. For instance a change in seasonality could have profound implications to overall hydrology if that change meant a state transition from snow to rain even with a relatively modest change in temperature. There's a robust literature (especially for the west coast) on the impacts of these projected changes. Maybe some incorporation of overall vulnerability across all these indices is warranted. Surely that's industry or stakeholder specific in what they deem "important" as highlighted in the case studies, but better discussion here may be warranted.

Author Response 20) We can add further discussion about the implications of elevated

vulnerability in a revision. This, combined with some of the previously proposed revisions, would strengthen the manuscript.

Specific Comment 21 (Section 4: Discussion): Along those same lines, you dedicate a lot of space both in terms of figures and text towards changes in seasonality (Figure 4-5) and FMI (Figures 2-3). Some discussion on whether you expect those to be the most consequential HL metrics in this region would be useful.

Author Response 21) We can further discuss changes in seasonality and FMI in a revision. These two metrics integrate numerous aspects of climate change into a single metric, and tend to be metrics that are of general interest. Thank you for pointing out a potential imbalance in our analyses.

———————————————————

---

## Author Response (AR1)

November 30, 2020

To:      HESS Editor

From:    Chas Jones, PhD

Subject:   Reconciliation of manuscript by Jones et al. (hess-2019-638)

Thank you for offering us the opportunity to respond to the reviewers' comments and feedback on our manuscript titled "Using hydrologic landscape classification and climatic time series to assess hydrologic vulnerability of the Western U.S. to climate". The manuscript was reviewed by one reviewer (RC2) that recommended that the paper be "revised and resubmitted". A second reviewer (RC1) suggested changing the manuscript into a technical note. We have addressed the specific concerns in the attached revision that improves the manuscript and makes it worthy of publication in HESS. We found the reviewer feedback to be insightful and it benefitted the manuscript. Attached you will find a copy of our response to the two reviewer comments and one short comment

*Please note that the reviewer line numbers reference the original manuscript submitted to HESS, while the response line numbers refer to the unmarked-up version of the revised manuscript.*

**Reviewer Comments (Submitted as RC2)**

1) General Comment

This paper aims to apply the hydrologic landscape approach to chronicling changes to the western U.S. under climate change. Using a vulnerability index, the authors aim to highlight locations more or less prone to changes in climate for various indices. It is apparent that the manuscript was a technically challenging effort to reconcile multiple datasets and climate scenarios and synthesize them in a GIS framework. The work there should not be discounted. The manuscript is generally well written but feels disjointed and an attempt to reconcile several disparate research efforts between a discussion on climate change, hydrologic landscapes, and vulnerability of socially/economically valuable locations. I understand what the authors are aiming for. If the authors aim to show all three aspects, they could be better unified. Overall, I think this paper is worthy of publication and the data analysis is commendable, but better structure and explanation is needed. I recommend the paper be revised and resubmitted.

> *Response)  Thank you for this feedback and recognition of the value of the research effort. RC2 recognizes that a strength of the manuscript is our attempt to integrate the fields of 1) climate change, 2) Hydrologic Landscape classification, and 3) the socioeconomic impacts of climate change. We highlight this strength of the analysis by adding/modifying the introduction which helped to emphasize these aspects of the study (L53-58; L138-145; added Section 4.2; and Section 5).  We also added language of this unifying concept into the Introduction and then revisit the unifying concept in the closing paragraphs of the Discussion or Conclusion sections.*

2) Specific Comment (Lines 64-65)

"The findings are consistent across studies in many areas of the globe...." How are they consistent across studies and which studies? And the second half of this sentence seems to contradict the first half when you say they aren't consistent without any citations being given. Which one is it?

*Response)  This sentence is referring to McAfee's 2013 study and was intended to summarize her findings. We clarified the text to indicate that they found that regional analyses were more inconsistent than national studies. (L68-70)*

3) Specific Comment (Line 175-177)

The methods describe that Leibowitz et al (2016) used a modification of Wigington et al (2013). It's a little unclear as to what the modification was. More clarity or explanation needed here.

*Response)  We added text to summarize Leibowitz et al. (2016)'s modification to the Wigington et al. (2013) methods (L190-196).*

4) Specific Comment (Section 2.3)

I'm a little unclear as to the selection of these dates and selection of data. Why is 1971-2000 considered the modern climate normal when such data is at least 20 years ago? It seems incongruous to have this be your ""modern"" normal when you consider "historical" data to be up to at least 2010 and state that the PRISM data you use for your calculation of modern normals goes from 1895-present. Why not have the modern normal represent a more recent time period?

*Response)  Thank you for pointing this out. We agree that the use of the term "modern" is inaccurate and we have removed it throughout the document. We chose to use the 1971-2000 period because the analysis complemented the Leibowitz et al. 2016 study, which used 1971-2000 as its defined "climate normal." We added explanatory text to our reasoning for defining our normal climate period as 1971-2000 (Section 2.3.1). We have also removed references to the "modern" climate normal throughout the manuscript.*

5) Specific Comment (Section 2.3; Lines 230-240)

I'm also unclear as to why monthly precipitation and mean temperature data is acceptable for the modern climate normal calculation (L230-232), but daily measurements are needed for the historical decadal analyses (L240) and they're subsequently averaged to monthly means anyways. The requirement for daily data caused you to employ a downscaling approach, potentially introducing more error. More explanation is needed here.

*Response)  While we alluded to this detail in the original manuscript, we added explanatory text that clarifies the reasons for these decisions (Section 2.3.2). As far as error from downscaling, while the 400m data clearly have greater resolution and less error than the 4km data, for the actual application these data were aggregated to assessment units with a mean area of 56 km². So, in practice, the larger 4km resolution of the downscaled historical analysis should still be appropriate for the scale of the assessment units.*

6) Specific Comment (Lines 262-265 and Fig. 2)

Better explanation is needed as to what were the criteria for choosing the 10 modeled emissions scenarios. Figure 2 appropriately shows their distribution in terms of precipitation and temperature, but how were the 10 out of the at least 38 chosen? Random draw? Some other selection criteria? Further, the coloring and subscript numbers in Figure 2 needs to be better explained in the caption.

*Response)  We now explain that we subjectively selected ten models that appeared to span the entire range of predicted climatic responses of the full ensemble in a distributed manner (L284-286). We also added clarifying information about the figure coloring and naming conventions of Fig. 2 to its caption.*

7) Specific Comment (Section 2.5; Lines 309-316)

Better explanation is needed for how these sites were selected and how their areal extent was decided (see Table 2). Site specific areal extent appears to range from 38 km2 to 4855 km2 (Table 2). Also, according to section 2.2 the target AU size is 80 km2 meaning that at the low end (e.g. Great Basin NP w/ area of 38 km2) is likely composed of a single AU and many only a few units. I get the challenges of making AUs a representative size across multiple different spatial datasets, but some discussion of how that AU size and differential location areal extent affects these location-based analyses is warranted.

*Response)  Specific sites were selected subjectively so that we could examine climate impacts at sites that may be of general interest. In addition, the range of Assessment Unit (AU) areas represents watersheds that are larger than hillslopes but smaller than large basins. We also explain that all of the AUs that had their centroid within the geographic boundary of a location were included in the AU analysis for each location. For instance, the Great Basin National Park (GBNP) was covered by a single AU, rather than numerous AUs because the centroids contained by other AU areas fell outside of the GBNP boundary. We added explanatory language to explain all of this background information (Section 2.5).*

8) Specific Comment (Lines 312-313)

"The time series for the decadal averages for each of the seven HL metrics..." I think you mean to say the seven climate related HL metrics here because things like elevation, subsurface permeability, and surface permeability aren't subject to change under this approach.

*Response)  Thank you for the good catch.  We added the 'climate-related' descriptor to our reference to the seven HL metrics (L341-342).*

9) Specific Comment (Lines 325-327)

The sentence beginning "In terms of the 1971-2000 climate normal period" needs some revision. I think it needs a clause saying, "followed by 24% of the area showing fall seasonality, 13% spring seasonality. "

*Response)  Absolutely. That was an awkward sentence that needed to be revised. The sentence structure has been improved as suggested (L354-356).*

10) Specific Comment (Lines 342-343)

Needs some clarification. What remaining models? You said in methods you only tested 10 and in the preceding text you said 3 may be wetter and 7 generally drier. What models are left?

*Response) Thank you for noticing that duplication of information that basically repeated the information with different wording. We have deleted the duplicative text and relocated this subsection to the supplemental materials per comment #21.*

11) Specific Comment (Line 355)

Several times in the results and discussion you point out patterns shown along major geographic features like mountain ranges (just as one example paragraph beginning on L351). It would be beneficial to show where those are like you do in the ecoregions in Figure 1. Many of the readers may be unfamiliar with where these features are so it makes it difficult to place the patterns you're describing.

*Response) In this paragraph, we do reference the White Mountains, which is Location #42 in Fig. 1 and table 2. We believe that we've only referenced place names that correlate to Locations identified in Table 2 or Fig. 1. We have added a reference to the assigned Location # when referencing place names (Fig. 1). We also refer to the "Sierra-Nevada Mountains", "Cascade Mountains", and "Mountainous areas in Idaho" [L461-463]. We improved this issue by either modifying these references throughout the manuscript.*

12) Specific Comment (Line 371)

Several times the authors talk about the sensitivity or vulnerability changes without talking about the direction of that change (but see sentence beginning on L371 "The map for S'" as an example). It'd be useful to make sure if you're saying an area is vulnerable to a change in climate, it's not just the metric (e.g. temperature) but also whether higher/lower or earlier/later.

*Response) While it is possible to talk about direction of change (higher or lower than the two standard deviations) for the projection of an individual climate model, the vulnerability index is the integration of ten individual models. It is possible for individual models to exceed the two standard deviation threshold from the mean in both the upper and lower directions, thus there is not a unique direction of change associated with our vulnerability index as we've defined it. We added text to the methods and results that clarifies this detail of our Vulnerability Index (L316-319, 364-368; Section 4.1).*

13) Specific Comment (Line 385-392)

I mentioned in my introduction that the paper was one that seemed disjointed between the paper being primarily one communicating climatic changes with a discussion of HLs and vulnerability thrown into the mix. This is an example of where the authors do a good job of uniting at least the HL approach with the climate indices. The HL approach is the story of the complete code where certain indices may play a more proximal role in given locations. This section does a good job of explaining how the changes in the climate indices could have differential effects based on things like elevation or the permeability metrics. More discussion like this is needed throughout the paper.

> *Response) Thank you for pointing out that the integration of the HL approach with the climate indices is a unique aspect of our manuscript and is worth expanding upon. We have added more text and additional HL context to our intro, discussion of climate and the associated socio-economic implications to the introduction, discussion, and/or conclusions. (L53-58; L138-139; L446-450; added section 4.2; and L538-540)*

14) Specific Comment (Figures 3 & 4)

These colors are hard to discern with much of the area looking a yellow color that according to the sale is no change. I wonder if more of a categorical variable would be appropriate here to show changes rather than a color ramp. The reddish hue is more noticeable in Figure 4 for sure, but I wonder if this could be better communicated.

> *Response) We appreciate the reviewers concern. However, Figures 3 and 4 do illustrate the actual geographic differences in FMI across large regions. Further, when mapping the differences categorically, the differences either appear exaggerated or absent. Thus, we prefer to retain these figures in their current form, as we consider categorical differences to be less accurate. However, we have pushed these results to the supplemental materials to shorten the manuscript and address reviewer comments.*

15) Specific Comment (Figure 7.1)

I wonder if here too classed variables may be better used to show variability by placing then in a low, medium, high type construct rather than a color ramp.

> *Response) We agree. We modified the images / legend as a classified variable.*

16) Specific Comment (Figure 8)

I'm confused by several aspects of this figure. Several of the figure panels don't have corresponding descriptions in the caption. For instance, there's a two separate panels for April 1 SWE and Snow, but the figure caption says "snow (April 1 SWE (mm))" as if they're combined somehow. This is confusing especially when the "Snow" panel has a y-axis labeled 1-2000 without any units. Also, the panel labeled "Climate" I believe is referenced as "FMI" in the caption. Also, on this panel the left y-axis is from -1 to 1 while the right axis is the categorical Arid-Very wet labels. This needs to be better explained as it's confusing even to someone familiar with HLs. Finally, the climate projection section is also confusing as sometimes it appears there's two lines while others there are several (e.g. Mt. Hood SWE panel). It may be cleaner to just show the high and low range lines rather than all the model scenarios I think you're showing. What the gray shaded area is showing also needs to be described in the caption.

> *Response) Thank you for the attention to detail. There was an error in the labeling of one of the figures. The "Snow" figure has been relabeled as "PET". We added clarification to the Climate / FMI panel of figures to address the reviewer's questions. At this time, we have chosen not to remove the red dashed lines that illustrate the individual climate model outputs.*

17) Specific Comment (Lines 426-521)

The discussion of the site-specific locations seems a bit disjointed in the discussion. I wonder if it'd be better served to be called out as case studies in a subsection. The discussion seems to go big picture, dive down to case studies, and then back out to a discussion of HLs. This organization seems a bit haphazard and cobbled together. I wonder if better flow and cohesiveness from section to section could be achieved here.

*Response) We added a separate subsection for the results and discussion of case studies (Section 4.3). We refined the language of "examples" to case studies throughout the document. This case study section was moved after the latter discussion of HLs for smoother continuity. Also updated the abstract and intro accordingly (L35-36; L138-145; Section 3.2.2; Section 4.3).*

18) Specific Comment (Sections 4 and 5: Results and Discussion)

The results section is dominated by description of the climatic time series and changes to the HL indices classifications, but explanation of those changes largely disappear in the discussion and is dominated by discussion of vulnerability. I get that the vulnerability index is an attempt to merge some of those ideas, but I would have expected better mixing of the climate and HL information in with the vulnerability discussion.

Response) *We have further highlighted the integration of climate, the HL classification approach, climate vulnerability, and socio-economic impacts throughout the paper, especially the intro, discussion, and conclusions. (Section 4.2; Section 4.3; Section 5)*

19) Specific Comment (Lines 260-262)

You stated in the methods that you chose from the highest emissions scenarios climate data projections (RCP 8.5). Better admonition of that fact needs to be detailed in the discussion as several other projection scenarios show lower degrees of change or better explanation of why you thought the high-end emissions scenarios were most representative needs to be explained.

*Response) We explain in the discussion that RCP 8.5 was selected because it most closely relates to the CO2 emission scenarios experienced to date (Section 2.3.3; Section 4.1).*

20) Specific Comment (Other)

I think there could be better discussion as to how having high vulnerability in a single metric could have profound implications in some areas while other areas may only be affected by having high vulnerability across multiple metrics. You get at some of this in the case study approach where certain grape varietals are more impacted by temperature changes say rather than precipitation changes, but I think that could be expressed better throughout including in the discussion of HLs. For instance, a change in seasonality could have profound implications to overall hydrology if that change meant a state transition from snow to rain even with a relatively modest change in temperature. There's a robust literature (especially for the west coast) on the impacts of these projected changes. Maybe some incorporation of overall vulnerability across all these indices is warranted. Surely that's industry or stakeholder specific in what they deem "important" as highlighted in the case studies, but better discussion here may be warranted.

> *Response)  We have added a discussion of single vs. multiple metric vulnerability (Section 4.1).*

21) Specific Comment (Section 4: Discussion)

Along those same lines, you dedicate a lot of space both in terms of figures and text towards changes in seasonality (Figure 4-5) and FMI (Figures 2-3). Some discussion on whether you expect those to be the most consequential HL metrics in this region would be useful.

> *Response)  Upon further reflection, we have decided to move these methods and results to the supplemental materials.*

**Reviewer Comments (submitted as RC1)**

1) General Comment

The manuscript, using existing indices and geospatial datasets, proposes a framework/rule-based decision making on the vulnerability of the Western U.S to future climate change. The manuscript is interesting and encompasses significant data management and GIS work. My general comments: Reading the manuscript, I have a feeling that HESS is not really the right journal for this work. Although interesting work, the manuscript seems to be a report/technical memorandum that is turned into a scientific manuscript. I would suggest this work may be better presented in other engineering or water management journals. This is just my recommendation on better presenting the work in its context to the right audience. Following that, it is rather difficult to provide a scientific feedback to this work. My feedback remains mostly on the clarification of presentation.

> *Response)      We are happy to see that the reviewer (RC1) found the manuscript interesting, although, as we describe below, we respectfully disagree with this reviewer's suggestion that this research would be more appropriate for an engineering or water management journal.  While the reviewer does provide specific feedback that is helpful for improving the manuscript, this feedback does not seem to justify the recommendation to submit to a different journal.  This critique also seems inconsistent with the feedback provided by the other reviewer, as well as the stated goals and scope of HESS) Nevertheless, we do find the specific comments provided by the reviewer to be valuable in helping us improve the manuscript, and we have done our best to address these in our revision.*
> *Response)*

2) General Comment

The use of English language is very good. The flow of the manuscript is smooth.

*Response)  Thank you.*

3)  General Comment

I am not sure if I really understand the linkage between the hydrological landscape classification and the current manuscript. As the authors mentioned in the introduction, the landscape classification is usually at finer resolution than catchment scale. What the author are doing, is more of clustering or zoning of possible system response to climate change (similar to hydrological modeling approach but with less hydrology as only indices are used). The AU are just a unit where the data is compiled at and this is not really linked to the sub-catchment variability intended landscape classification at catchment level.

*Response)  We have added clarification of this information and process (L121-145; L180).*

4)  General Comment

It seems the authors have a decision tree in mind that they use for classification using the input data. I would suggest the author to provide a schematic of their decision or algorithm that provide readers with better understanding of the method. Similarly, there is no visualization of the shapefile/regions used to create the vulnerability map.

*Response)  While this process is not a decision tree, we have created a new figure in attempt to summarize the HL code development, the use of historic and future climate projections to generate figures and the vulnerability maps (Figure 2). In addition, the HL classification specifics are described in the methods: Climate (Section 2.2.1); Seasonality (Section 2.2.2); Subsurface Permeability (Section 2.2.3); Terrain (Section 2.2.4); and Surface Permeability (Section 2.2.5). We further clarify that the vulnerability maps depict the ~24,000 AUs that were classified for each Vulnerability parameter, since it wouldn't be very helpful to create a map that shows that level of detail.*

5)  General Comment

I would say the context of vulnerability is missing here. What is it used for? What is the intended motivation behind this vulnerability assessment?

Response)  *The context of vulnerability is woven into the entire manuscript and the abstract. We specifically discuss how vulnerability is used in our assessment in L133-138.  We also added some introductory language to the end of the first paragraph (L53-58).*

6)  General Comment (Section 3: Results)

The result section is presented very quickly in (few) paragraph(s).

*Response)  We believe this material adequately communicates the results of our study when combined with the tables and figures. The text portion of our results equates to*

*1576 / 8125 words (19% of the text of the primary manuscript), which does not include tables or figures that will be placed within the section.*

7) General Comment (Section 4: Discussion)

The discussion is kind of back to front. It is rather wordy. I would say it can be significantly shorter and focused on the interpretation of the results given the aim of this study.

Response)

*We streamlined the content of the discussion and look for opportunities be more brief, while maintaining clarity. We balanced this comment with those by the other reviewer, who suggested adding a subsection and expanding the discussion. However, we restructured the discussion so that the case studies have been moved to the end, so hopefully that helps address the sense that the Discussion had been presented back to front. The discussion is now 1583/8125 words (19% of the paper (inclusive of Intro through conclusions)).*

8) General Comment (Section 5: Conclusions)

Conclusion session is very vague. I would suggest the authors to come up with few bullet points Conclusions which readers can have as take-home message. Also, the discussion, my pervious comment, can evolve along the line of the conclusion (I mean bullet point conclusions can help discussion significantly).

*Response) We emphasize our intended take home messages in the conclusions (Section 5)*

9) General Comment

My overall suggestion is to change the manuscript into technical note. I would strongly suggest shortening of the manuscript and remove wordy sections (for example, in discussion). Explain the decision tree visually and elaborate that in methodology section. Present the forcing and geospatial data in the decision tree and also visually. I believe major revision is inevitable.

*Response) We respectfully disagree with the reviewer's recommendation to change the manuscript into a technical note; see response to comment #1. We shortened the discussion section (while adding a subsection to it) from 1615 to 1583 words. We have attempted to shorten the overall manuscript (now 8125 words) and have removed sections from the results and pushed those to the supplemental materials. While we do not agree that a decision tree is the proper graphic to summarize our overall process, we have added Fig. 2 to illustrate our overall research process.*

**Short Comment (submitted as SC1)**

1) I noted that you cited my work describing vulnerability of stream temperatures to climate change. This paper by Sulochan Dhungel and others may be as relevant to this study since they looked specifically at hydrologic changes in response to climate change: https://onlinelibrary.wiley.com/doi/abs/10.1002/rra.3029

*Response) Thank you. We have reviewed the paper and now cite it at L76-78.*

[revised manuscript text omitted]

Supplemental Material

[Figure]

Figure S1. Hydrologic Landscape maps of the United States that were used in the HLVA analysis [(a) Subsurface
Permeability, (b) Seasonality of precipitation surplus, (c). Surface permeability, (d) Climate, and (e) Terrain]. Notes: The
seasonality map for the PNW has been updated from the original Leibowitz 2016 HL map, as we separated their winter
seasonality into two seasons (winter and fall). In addition, the subsurface permeability maps were only completed for the
western most portions of the U.S.

[Figure]

[Figure]

**Figure S2. Decadal change in Feddema Moisture Index relative to 1971–2000 normal period. Red and blue colors indicate drier and wetter average conditions than 1971–2000, respectively.**

**Figure S3. Projected change in Feddema Moisture Index for 2041–2070 relative to 1971–2000 for ten climate models. Red and blue colors indicate drier and wetter conditions than the 1971–2000 base period, respectively. Abbreviated model names correlate to those in Table 1.**

[Figure]

**Figure S4. Decadal change in seasonality of water surplus since 1901 relative to 1971–2000. Red and blue colors indicate**
**earlier and later seasonality than the 1971–2000 base period, respectively.**

[Figure]

Figure S5. Projected change in seasonality of water surplus for 2041–2070 relative to 1971–2000 for ten climate models.
Red and blue colors indicate earlier and later seasonality than the 1971–2000 base period, respectively. Abbreviated model
names correlate to those in Table 1.

[Figure]

**Figure S6. Vulnerability indices for temperature, precipitation, potential evapotranspiration, snow water equivalent (April**
**1), S' (available water), Feddema Moisture Index, and seasonality. The least vulnerable locations are those projected to be**

---

## Author Response (AR2)

March 26, 2020

To:     HESS Editor

From:   Chas Jones

Subject:   Reconciliation of manuscript by Jones et al. (hess-2019-638)

Thank you for offering us the opportunity to respond to the reviewers' comments and feedback on our manuscript titled "Using hydrologic landscape classification and climatic time series to assess hydrologic vulnerability of the Western U.S. to climate". The manuscript was reviewed by two reviewers that recommended acceptance with minor edits. We have addressed the specific concerns in the attached revision that improves the manuscript and makes it worthy of publication in HESS. Both rounds of reviewer feedback were insightful and benefitted the manuscript. Attached you will find a copy of our response to the two reviewer comments.

**Reviewer #2**

1) General Comment

   The authors here should be commended for a marked improvement in this revised paper. The revised organization and structure help guide the reader through the study better with a clearer sense of overall purpose and direction. I appreciate the thoughtful response to comments and the commitment made to addressing previous comments and concerns I had. I recommend acceptance of this paper with a few minor points that the authors may want to consider.
   *Response)*     *Thank you.*

2) I'd recommend the authors re-read this paper a final time for grammar and sentence structure and minor edits. Overall, the paper reads well, but there are a few places where a word seems missing. For example:

   L49: I think the authors mean to add a "to" so it reads "and is related TO the threats of increased flooding.
   *Response)*     *Thank you.  This has been addressed.*
   L460: the sentence here ends with the verb "are". I'd revise so it doesn't end that way.
   *Response)*     *Thank you.  This has been addressed.*

3) L60-82. This paragraph reads a bit like a cascade of a literature review. Maybe edit this paragraph to highlight common trends or take-home messages rather than Researcher X found this and Researcher Y found that. It provides good information but could be synthesized better.
   *Response)*     *Modified to provide better synthesis and added a summary sentence.*

4) Section 2.2.1 and by relation 2.2.2. For all the other HL indices you include the relevant dataset used to characterize that variable whereas we don't find out about the climate datasets till later in section 2.3. Maybe at least reference that the datasets used to develop those indices will be discussed in a follow-on section?
   *Response)*     *In line 206, we reference Section 2.3 as the location that we will describe the climate datasets, which I believe is what the reviewer is requesting.  Thus, we didn't make any edits in response to this comment.*

5) L272-275. Was there a comparison done as to the degree of match between the climate normals and the historical climate analyses for overlapping years? The addition of L266-271 adds valuable context, but there any comparison done to how the downscaled 4k resolution dataset ended up matching (or not matching) the 400m high resolution one?

Response)        *Good question. For the overlapping timeframe of 1971-2000, both the 4k and 400m data were used in the downscaling process so that the error for the rescaled data was essentially non-existent.  We did not have the high resolution 400m resolution data available for any individual month and year between 1971 to 2000, as we only had the monthly 30 year normal available at 400m. Therefore, there is insufficient information to present for a comparison.*

6) L314-320. I appreciated the additional language and consistent reference to your vulnerability evaluation not being indicative of a direction of change but rather a 2 SD threshold in either direction. It makes it much clearer. Based on Fig 5, it appears that precipitation and S' are particularly sensitive to going in either direction. Was there any evaluation done to discern those instances when it could go either direction? That seems as if it could point to less certainty in the results although it fits into your construct of a marked change of +/- 2 SD. Perhaps add a sentence or two as to implications of going in either direction?

Response)        *"Uncertainty" doesn't feel like the proper term given that these projections in vulnerability were defined by specific criteria. Analyzing "variability" in modeled projections seems more appropriate than uncertainty in this context. The Reviewer references Figure 5 as providing sufficient information to indicate more 'uncertainty' in two parameters, but Fig. 5 only provides 3 case study examples and may not be appropriate for drawing general conclusions. Figure A2 illustrates a broader view of the same information than Figure 5, but even then, Figure A2 depicts only 42 specific locations. The authors believe that it is more appropriate to examine the spatial uniformity of our vulnerability assessment presented in Figures 4 and S6, which provides the same information in a spatial context that provides more information about how consistent various geographies are to the analysis. Those figures combined with our comparison of various EPA Level 2 ecoregions was an attempt to see if the variability in HL vulnerabilities were correlated with specific geographic or ecological characteristics.  In our analysis, we discussed the similarities in geographic response that correlates with various ecoregions in regards to precipitation (lines 369-373), S' (lines 378-380), and FMI (lines380-384), all of which were found to have more spatial or climatological variability relative to the other hydrologic parameters.*

7) L394. Maybe add (SWE) after snow accumulation so it's clear what the text is referring to in Table 4.

Response)        *Thank you.  This has been addressed.*

8) L422. Maybe put S' at the end of this list so it's parallel with how the variables are shown in the Figure.

Response)        *Thank you.  This has been addressed.*

9) Table 2. You might consider making this an appendix or supplemental table. Maybe it'll take up less space in a published version, but in my print-out it currently covers three complete pages.

Response)        *Thank you. We think that this table will be formatted to take up less space in the edited journal. We would prefer to keep it within the manuscript rather than as an appendix but will reconsider if the formatting does not work out.*

10) Table 4. Again, likely just a formatting issue, but the percentages for 3 of the 5 indices are shown across two lines. This should be on a single line as the subsurface permeability column shows. For instance, climate column and temperature row says 70 (next line) %, should just be 70%.

*Response)* *The standards for this journal request that there be a "space" between the number and the % symbol. I assume that the copy editors will format the table appropriately, but if there are any issues, we will address them in the formatting stage with the editors. Thanks!*

11) Figure 2. Panels B and C and the legend for Panel E is illegible in the printed-out version. Again, may be improved in a published version, but I'd increase font sizes so they can be seen. I don't think anything below the font size shown for the labels in Panel A will work. Maybe make the figure landscape to give yourself more space?

*Response)* *We are happy to work with the editors as needed to ensure that our graphics are legible in the printed journal. We are also flexible on whether the graphic is presented in a landscape or vertical orientation.*

**Reviewer #3**

1) The paper presents an interesting and simple methodology to assess "vulnerability" of landscapes to climate change. The contribution is technically correct, I just would have liked to see more discussion on the lessons learned from developing and applying the model. A couple of lines on the authors' thoughts about how to analyze the results for the case of other industries (similar to the analysis of the two grape varieties in the example) would be great.

*Response)* *Great idea. We have added a couple of sentences to the Summary and Conclusions section (lines 545-559).*